# A fibroblast-like endothelial cell state promotes atherosclerosis via C/EBPβ-activated TGF-β signaling

Linge Fan[1,4], Yingyi Zhu [1,4], Yi Li[1], Zixin Ji [1], Kefan Ma[1], Ying Zhang[1], Leiting Wei[1], Junbo Chen[1], Yuanqing Jiang[1], Dongwu Lai[1], Lingfeng Qin[2], Guosheng Fu [1], Michael Simons [2], Liang Xu [1], Luyang Yu [1,3 ✉] & Cong Qiu [1,3 ✉]

## Abstract

**Endothelial cell (EC) dysfunction is a critical driver of chronic vascular inflammation and atherosclerosis. However, the molecular details of EC state dynamics during vascular disease progression remain ill-defined. Here, we used in-depth single-cell RNA sequencing to map transcriptional landscapes and molecular signatures of EC phenotypic plasticity during atherosclerosis in the mouse arota. This analysis identified a unique fibroblast-like EC population in atherosclerotic blood vessels, characterized by high expression of endothelial activation markers and extracellular matrix (ECM) remodeling, which increased with disease severity. Pseudotime trajectory analysis revealed that these fibroblast-like ECs represent terminal states of endothelial-mesenchymal transition (EndMT) during atherosclerosis. Further, the transcription factor C/EBPβ was identified as prominent driver of this phenotype transition as evidenced in vivo and in vitro. Mechanistically, inflammatory cytokines induce C/EBPβ, triggering TGF-β signaling and subsequent regulation of downstream genes via upregulation of TGF-β receptor type I (TGFBR1) through direct interaction with its promoter. Endothelial overexpression of C/EBPβ in vivo exacerbated atherosclerotic plaques, increased vascular inflammation, and elevated endothelial TGFBR1 levels. These findings highlight endothelial C/EBPβ as a novel regulator of TGF-β signaling and pathological fibroblast-like EC phenotypes during atherosclerosis, linking cytokine-driven inflammation with TGF-β-mediated endothelial dysfunction.**

**Keywords** Atherosclerosis; C/EBPβ; Endothelial Cell Plasticity; Single-cell RNA Sequencing; TGFBR1
**Subject Categories** Cell Adhesion, Polarity & Cytoskeleton; Immunology; Vascular Biology & Angiogenesis

## Introduction

Atherosclerotic cardiovascular diseases, involving myocardial infarction, ischemic heart disease, stroke, and peripheral arterial disease, are the leading cause of death worldwide (Dai et al, 2022). Atherosclerosis, characterized by chronic vascular inflammation, is a complex, progressively developing disease primarily initiated by hyperlipidemia and abnormal shear stress-induced vascular injury (Hansson et al, 2006; Libby et al, 2019). Currently, although clinical lipid-lowering drugs such as statins and PCSK9 inhibitors can significantly reduce cholesterol levels, these treatments have not been sufficient to prevent the rising incidence of atherosclerotic major cardiovascular events (Chen et al, 2020). This highlights the need to define novel molecular events responsible for atherosclerosis progression (Libby, 2021; Libby et al, 2016).

Endothelial cells (ECs) are vital components of blood vessels, playing critical roles in cardiovascular homeostasis. They regulate blood fluidity, control vascular tone, influence angiogenesis, modulate monocyte/leukocyte adhesion, and affect platelet aggregation (Sun et al, 2019). During the inflammatory process induced by pro-atherogenic risk factors such as oxidized low-density lipoprotein (Ox-LDL), pro-inflammatory cytokines or oscillatory shear stress, partial ECs become dysfunctional and promote the attachment and transmigration of immune cells to the vascular wall, thereby initiating and promoting atherosclerosis (Gimbrone and Garcia-Cardena, 2016; Medina-Leyte et al, 2021; Sun et al, 2019; Wang et al, 2022). Although EC heterogeneity in atherosclerosis has been proven, the transcriptional drivers of the pathological EC phenotype and their contribution to disease progression remain poorly defined.

A hallmark of atherosclerotic EC dysfunction is endothelial-to-mesenchymal transition (EndMT), a process marked by the loss of endothelial identity and acquisition of mesenchymal features, including extracellular matrix (ECM) remodeling and inflammation (Chen et al, 2015; Dejana et al, 2017; Souilhol et al, 2018; Yurdagul et al, 2016). Notably, EndMT-derived fibroblast-like cells

[1]Key Laboratory of Cardiovascular Intervention and Regenerative Medicine of Zhejiang Province of Sir Run Run Shaw Hospital, MOE Laboratory of Biosystems Homeostasis & Protection, College of Life Sciences, Zhejiang University, Hangzhou, Zhejiang 310058, China. [2]Cardiovascular Research Center, Interdepartmental Program in Vascular Biology and Therapeutics, Department of Internal Medicine, Yale University School of Medicine, New Haven, Connecticut 06520, USA. [3]Institute of Cancer, Zhejiang University-Lishui Joint Innovation Center for Life and Health, Zhejiang University, Hangzhou 310058, China. [4]These authors contributed equally: Linge Fan, Yingyi Zhu. ✉E-mail: luyangyu@zju.edu.cn; congqiu@zju.edu.cn

represent a significant source of mesenchymal cells in atherosclerotic plaques (de Winther et al, 2023; Evrard et al, 2016), yet the functional relevance, molecular signature, and transcriptional mechanisms driving this pathological phenotype in response to pro-atherogenic stimuli remain underexplored.

Recent advances in single-cell RNA sequencing (scRNA-seq) permit a deep and comprehensive description of the heterogeneity of ECs involved in arterial vasculature at a single-cell resolution (Chen et al, 2019; Cheng et al, 2023; Zhao et al, 2021). Therefore, by comparing the genetic profiles of functional and dysfunctional ECs in normal versus atherosclerotic aortas, we may uncover the gene profiles of fibroblast-like ECs and elucidate new pathogenic mechanisms underlying vascular inflammation in atherosclerosis.

Here, we combined scRNA-seq, functional genomics, and in vivo models to dissect EC heterogeneity and identify novel drivers of endothelial dysfunction in atherosclerosis. We report the discovery of a fibroblast-like EC population that emerges during disease progression, characterized by a pro-inflammatory, ECM remodeling phenotype. Further investigation uncovered that the transcription factor C/EBPβ, induced by inflammatory cytokines, promotes this pathological endothelial phenotype by activating the TGF-β signaling pathway. Specifically, C/EBPβ functions as a novel transcriptional regulator of TGFBR1, directly upregulating its expression by binding to the TGFBR1 promoter. C/EBPB overexpression at the endothelium promoted endothelium inflammation and atherosclerosis in vivo. Our findings not only advance the understanding of EC plasticity in atherosclerosis but also suggest the C/EBPB-TGFBR1 axis as a novel therapeutic target for atherosclerotic disease.

# Results

## scRNA-seq reveals distinct endothelial cell populations in atherosclerotic mouse aortas

To comprehensively characterize the molecular heterogeneity of aortic ECs during atherosclerosis, we performed the scRNA-seq analysis of ECs from three distinct groups: normal aortas of C57BL/6 mice fed a standard diet; mild atherosclerotic aortas of $Apoe^{-/-}$ mice fed a standard diet; and severe atherosclerotic aortas of $Apoe^{-/-}$ mice fed a high-fat diet. After feeding, plasma triglyceride, total cholesterol, and LDL-C were significantly increased in both $Apoe^{-/-}$ groups when compared to the C57BL/6 mice, and the group with a high-fat diet showed the highest plasma LDL and total cholesterol levels (Fig. EV1A–C). Atherosclerotic lesions were only observed in $Apoe^{-/-}$ groups, with lesion severity correlating with plasma lipid accumulation (Fig. EV1D,E). Following enzymatic digestion and the removal of perivascular fat, live ECs were collected by CD45$^-$CD31$^+$ sorting via fluorescence-activated cell sorting (FACS, Fig. EV2) and were subsequently barcoded and sequenced using a 10x Genomics platform (Fig. 1A). The mean reads per cell were 262,296 for the normal control group, 110,061 for the mild group, and 104,957 for the severe group, which corresponds to a median of 2440 genes per cell after pre-normalization.

Unbiased clustering of the integrated datasets using t-distributed stochastic neighbor embedding (t-SNE) revealed 14 distinct EC clusters (Fig. 1B–D). The high level of Pecam1 expression confirmed the endothelial identity of these clusters (Fig. 1C), while

minimal Ptprc (CD45) expression excluded hematopoietic contamination (Fig. EV3A). Differentially expressed gene (DEG) analysis identified cluster-specific markers, with the top ten DEGs for each cluster visualized in a heatmap (Fig. EV3B; Table EV1). In total, 446, 755, and 1302 cells were included for subsequent analysis after quality control from the normal, mild, and severe groups, respectively.

Next, clusters 10–13 were excluded due to minimal Pecam1 expression, featured expression of smooth muscle cell (SMC) markers (Myh11, Acta2, and Tagln), or low cell counts (Fig. EV3C–G; Table EV2). Re-clustering of the remaining 10 EC populations (clusters 0–9) highlighted unique transcriptional programs (Fig. 1E,F), consistent with previous reports on aortic EC heterogeneity (Kalluri et al, 2019).

## Characterization of the unique fibroblast-like EC population in atherosclerotic aortas

Notably, cluster 2 exhibited the pronounced expression of fibroblast-associated genes (Lum, Dcn), which were absent or minimally expressed in other clusters (Fig. 2A). This unique gene expression profile defined a fibroblast-like EC subset, which we hypothesized to play a role in atherosclerotic progression. To further dissect the pathological relevance of these cells, we analyzed gene signatures linked to EC fate, EndMT, and inflammation. The fibroblast-like EC phenotype shows a broad increase in the expression of EndMT-related and inflammation-related genes, accompanied by a downregulation of EC fate markers (Fig. 2B). Moreover, Gene Ontology (GO) and Kyoto Encyclopedia of Genes and Genomes (KEGG) pathway enrichment indicated DEGs in these ECs enriched in the ECM structural constituent and organization, ECM-receptor interaction, and inflammation, that are closely associated with atherosclerosis (Figs. 2C and EV4).

To explore the dynamic evolution of these ECs during atherosclerosis, we performed pseudotime trajectory analysis on 1691 cells from six clusters (0, 1, 2, 4, 5, 9) based on their correlations (Fig. EV5). Points were identified, representing different cellular states or transitions within these clusters. Clusters 0 and 1 were positioned on the right side of the trajectory (Fig. 2D). The proportion of cluster 0 and cluster 1 ECs decreased as atherogenesis progressed (Table EV3). Additionally, these two clusters exhibited high expression levels of genes associated with plasma membrane, membrane, and cytoplasm GO enrichment terms, as well as KEGG pathways such as oxidative phosphorylation, metabolism of xenobiotics by cytochrome P450, and antigen processing and presentation (Fig. EV6). These findings indicate that the clusters 0 and 1 serve a crucial role in maintaining fundamental cellular functions, likely representing normal ECs, occupying the early state of the trajectory. In contrast, cluster 2 localized to the terminal branch of the trajectory (Fig. 2E), coinciding with the plotting of the markers of fibroblast, inflammation, and the ECM along the trajectory (Figs. 2F–I and EV7).

To further investigate the pathological role of the fibroblast-like EC subset in atherosclerosis, the expression levels of fibroblast marker genes (Serpine1, Fn1, Lum, and Dcn), endothelial activation marker genes (Icam1, Sele, Cxcl1, and Ccl2), and the ECM remodeling marker gene (Col8a1) were analyzed across the normal, mild, and severe groups. The results suggest that the expression of

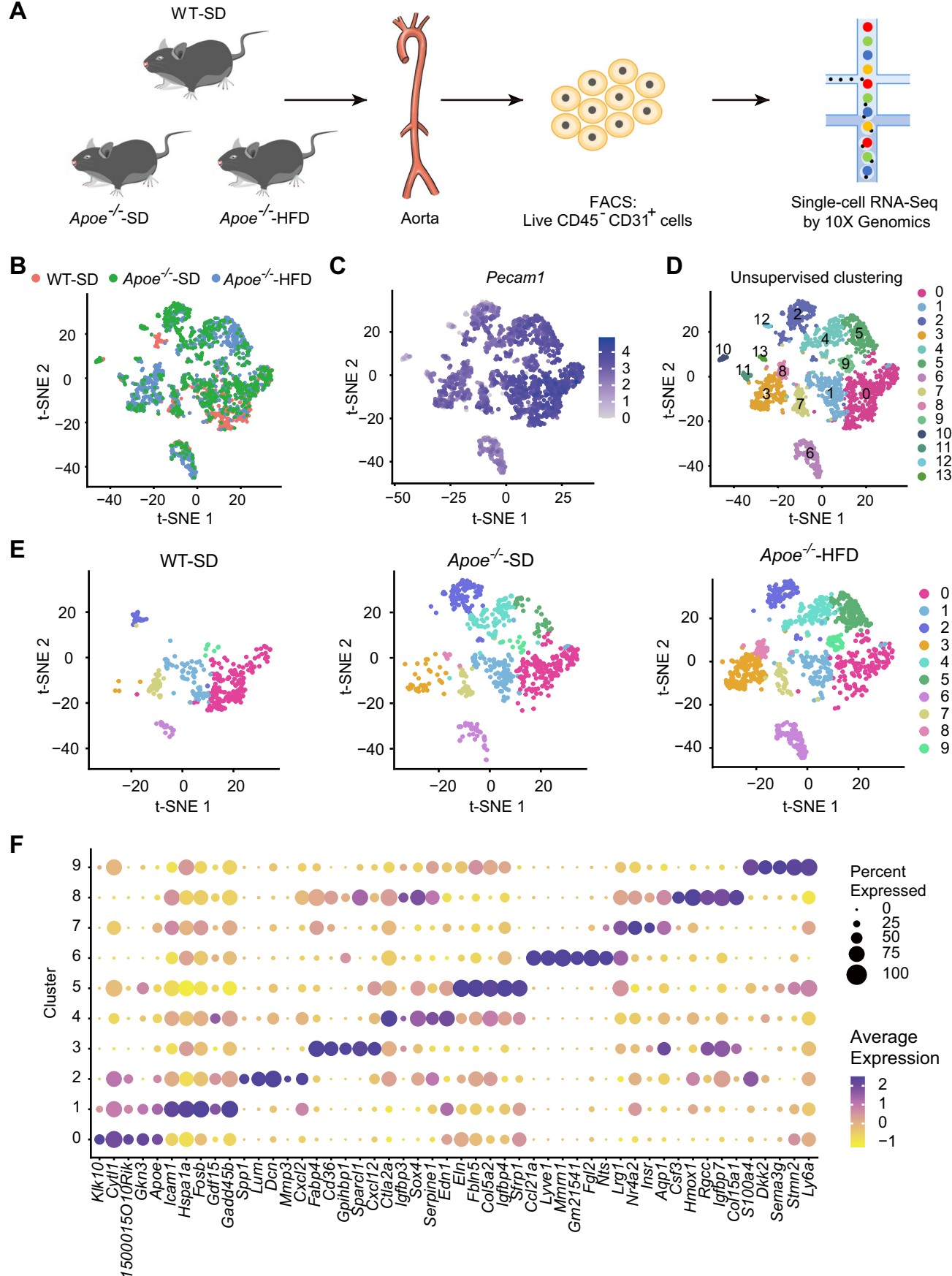

Figure 1. scRNA-seq reveals distinct EC populations in mouse aortas.

(A) Schematic diagram of the scRNA-seq procedure. WT wildtype, SD standard diet, HFD high-fat diet. Cells from each group were pooled from four mice. (B) t-SNE plot of ECs from three groups, with colors denoting different groups. (C) Feature plots showing relative expression levels of *Pecam1* in all captured cells from all groups. (D) t-SNE plot of ECs colored by cluster number. (E) t-SNE plot of the ten selected EC clusters for the three experimental groups, with colors denoting cluster number. (F) Dot plot showing average expression and percent expressing cells for the top five marker genes of the ten selected clusters (clusters 0–9). Source data are available online for this figure.

these genes was higher in atherosclerotic groups when compared to the normal control group (Fig. 2J), indicating a potential association between fibroblast-like ECs and atherogenesis. Additionally, the immunostaining of human atherosclerotic coronary arteries demonstrated elevated LUM expression in plaque-associated endothelium (Fig. 2K), supporting the presence of fibroblast-like ECs in atherosclerotic lesions. Collectively, these findings suggest a potential role for fibroblast-like ECs in atherosclerosis, thereby highlighting the need for further investigation into their functional significance.

## C/EBPβ functions as a master regulator of the fibroblast-like EC phenotype

Next, we sought to unravel the transcription regulatory mechanism of the fibroblast-like EC phenotype. We compared the expression of all 827 mouse transcription factors saved in the TRRUST database (Han et al, 2018) across the six clusters (0, 1, 2, 4, 5, and 9). Cluster 2 exhibited significant upregulation of 53 transcription factors (log2 fold change ≥1.2 when compared to clusters 0/1), with *Cebpb* ranking as the most highly upregulated (Fig. 3A,B). In addition, endothelial *Cebpb* expression increased with atherosclerosis severity (Fig. 3C). Moreover, the immunostaining of human atherosclerotic coronary arteries and mouse aorta validated elevated C/EBPβ expression in plaque-associated endothelium (Fig. 3D–G), thus confirming the positive role of endothelial C/EBPβ in promoting atherosclerosis. Moreover, the in vitro stimulation of human aortic endothelial cells (HAECs) with the pro-inflammatory cytokines IL-1β and TNF-α (Poli, 1998) induced C/EBPβ expression at both the mRNA and protein levels (Fig. 3H–J), thus implicating inflammatory signaling in its activation during atherogenesis.

An RNA-seq performed with three biological replicates from HAECs with or without adeno-C/EBPβ (Ad-C/EBPβ) virus infection revealed the significant upregulation of mesenchymal markers, such as *DCN*, *LUM*, *SERPINE1*, *FN1*, and *COL8A1*, as well as inflammatory markers, such as adhesion molecules, chemokines, and cytokines (Fig. 4A). GO analysis of DEGs highlighted enrichment in biological processes related to EC fate, inflammatory response, cell adhesion, chemotaxis, collagen-containing ECM, and protein binding that are critical for the atherosclerosis process (Fig. 4B). To further validate the pathological role of C/EBPβ in ECs, we generated CEBPB-knockout (KO) ECs using the CRISPR-Cas9 system (Fig. EV8). Following the verification of KO efficiency, the effects of C/EBPβ deficiency on the expression of key mesenchymal and inflammatory markers, cell morphology, and EC adhesion to monocytes were determined. Notably, C/EBPβ KO significantly attenuated the cytokine-induced upregulation of key mesenchymal and inflammatory markers at the mRNA level (Fig. 4C–H). Morphologically, cytokine-stimulated HAECs exhibited fibroblast-like characterization with an increased length-to-width (L/W) ratio, a phenotype abolished in C/EBPβ KO cells (Fig. 4I,J). Furthermore, functional assays demonstrated that C/EBPβ deficiency markedly reduced monocyte (THP-1) adhesion to the endothelial monolayer (Fig. 4K,L).

To further confirm the pathological role of C/EBPβ, we overexpressed C/EBPβ in HAECs. Consistent with the loss-of-function phenotype, C/EBPβ overexpression significantly enhanced the expression of key mesenchymal and inflammatory markers (Figs. 4M–R and EV9A–E). Morphologically, HAECs with C/EBPβ overexpression exhibited fibroblast-like characteristics with increased L/W ratio (Fig. 4S,T). Moreover, functional assays demonstrated that C/EBPβ overexpression enhanced monocyte (THP-1 cell) adhesion to the endothelial monolayer (Fig. 4U,V) and impaired endothelial barrier function, as evidenced by increased FITC-dextran permeability (Fig. EV9F).

Taken together, these data suggest that endothelial C/EBPβ functions as a pivotal inducer of EC dysfunction during atherogenesis.

## C/EBPβ activates TGF-β signaling by directly upregulating TGFBR1 expression

Next, we aimed to unravel the molecular mechanisms underlying CEBPB-mediated EC dysfunction. Gene set enrichment analysis (GSEA) from RNA-seq indicated enrichment in the TGF-β signaling pathway, which is the main driver for EndMT (Fig. 5A) (Chen et al, 2019; van Meeteren and ten Dijke, 2012). To further test whether C/EBPβ enhances TGF-β signaling, p-SMAD2, the marker for TGF-β signaling activation, was determined. Immunoblotting showed that Ad-C/EBPβ significantly enhanced p-SMAD2 in HAECs (Fig. 5B).

To test whether C/EBPβ-induced phenotypic changes were dependent on TGF-β signaling, we treated Ad-C/EBPβ HAECs with the TGF-β receptor inhibitor SB431542. mRNA analysis indicated that SB431542 treatment reduced the elevation of mesenchymal markers as well as inflammatory markers induced by C/EBPβ (Fig. EV10A–H). In addition, the inhibitor restored endothelial barrier function and reduced monocyte adhesion induced by Ad-C/EBPβ (Fig. EV10I–K). These results indicate that C/EBPβ may promote a pro-atherogenic phenotype by activating TGF-β signaling. Since SB431542 inhibits TGF-β signaling by suppressing ACVR1B, TGFBR1, and ACVR1C, we subsequently analyzed the fold change of *ACVR1B*, *TGFBR1*, and *ACVR1C* from RNA-seq and found that *TGFBR1* was the most significantly upregulated receptor (Table EV4). Thus, we determined TGFBR1 expression after C/EBPβ overexpression. As shown in Fig. 5C–E, Ad-CEBPB improves *TGFBR1* mRNA and protein levels, indicating that C/EBPβ may activate TGF-β signaling by increasing TGFBR1 expression. To determine whether TGFBR1 is required for C/EBPβ-

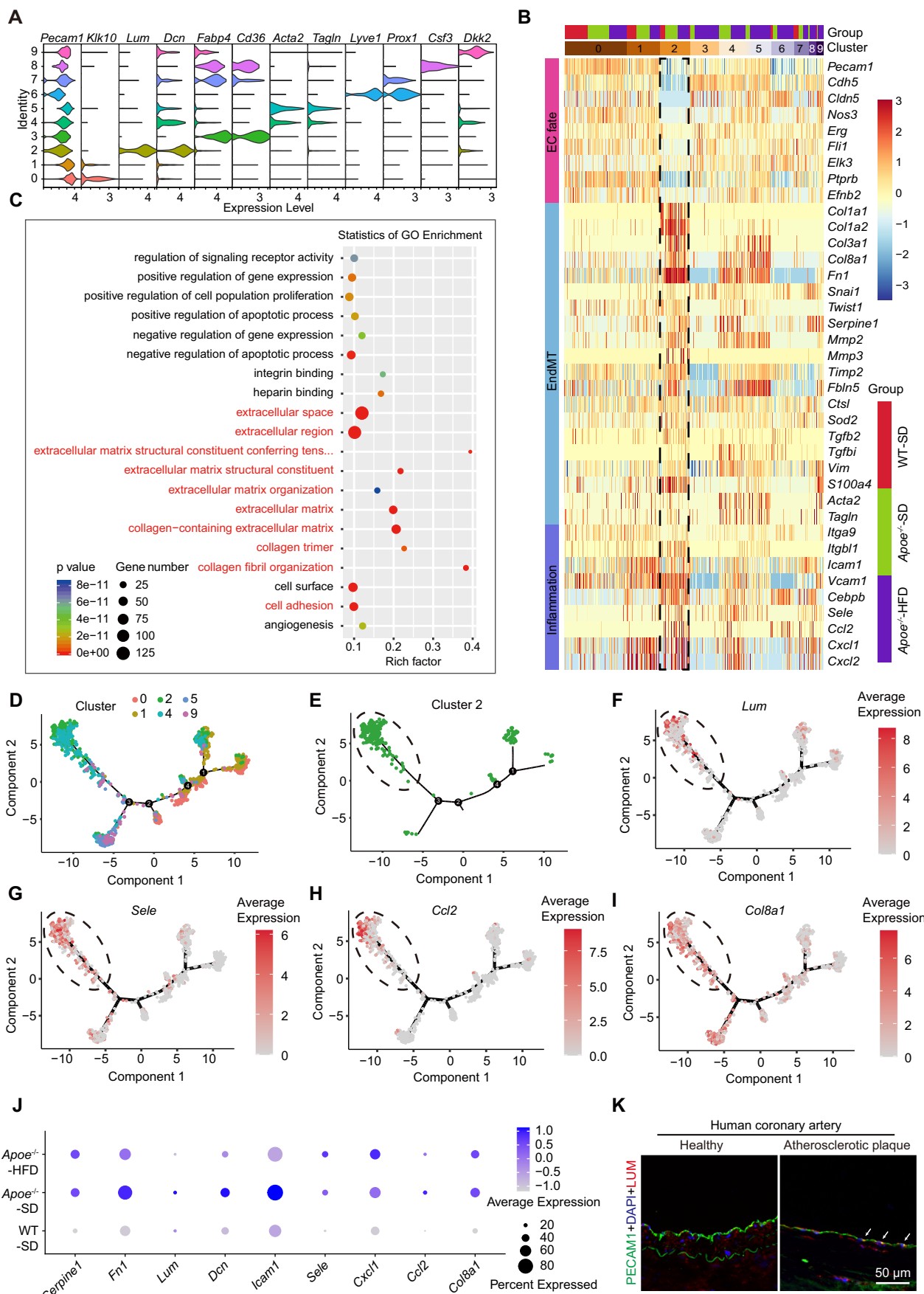

◄

**Figure 2. Characterization of fibroblast-like ECs.**

(A) Violin plot revealing the expression patterns of *Pecam1*, *Klk10*, *Lum*, *Dcn*, *Fabp4*, *Cd36*, *Eln*, *Acta2*, *Tagln*, *Lyve1*, *Prox1*, *Csf3*, and Dkk2 across clusters 0–9. (B) Heatmap displaying related gene expression patterns arranged by cluster and group. The gene expression pattern of cluster 2 is indicated by a black dashed box. Data analysis was performed from 423 cells (WT-SD group), 705 cells (*Apoe*⁻/⁻-SD group), and 1245 cells (*Apoe*⁻/⁻-HFD group). All gene expression profiles were preprocessed using zero-centering normalization to facilitate comparative analysis. (C) Dot plot of the top 20 GO terms enriched in the DEGs of cluster 2 compared to other clusters. Red text indicates genes enriched in ECM-related items. (D) Monocle pseudotime trajectory analysis of ECs from six clusters, as indicated. (E) Cluster 2 located at the terminal branch of the trajectory, indicated by a black dashed ellipse. (F–I) Expression patterns of *Lum* (F), *Sele* (G), *Ccl2* (H), and *Col8a1* (I) along the pseudotime trajectory. (J) Dot plot showing average expression and proportion of expressing cells for representative fibroblast, inflammation, and ECM-related genes in ECs from normal aortas (WT-SD), mild atherosclerotic aortas (*Apoe*⁻/⁻-SD), and severe atherosclerotic aortas (*Apoe*⁻/⁻-HFD). (K) Representative IF staining of PECAM1 (green), LUM (red), and DAPI (blue) for healthy and plaque-associated endothelium in human coronary arteries (*n* = 4). White arrows indicate PECAM1/LUM colocalization. Scale bar, 50 µm. Source data are available online for this figure.

mediated TGF-β signaling, we knocked down *TGFBR1* in Ad-C/EBPβ-infected HAECs. As expected, the loss of *TGFBR1* inhibited C/EBPβ-induced SMAD2 phosphorylation (Fig. 5F) and reduced the mesenchymal and inflammatory genes induced by C/EBPβ (Fig. 5G–Q). These data suggest that C/EBPβ promotes a pro-atherogenic phenotype by activating TGF-β signaling through TGFBR1 upregulation.

The above findings highlight C/EBPβ as a potential transcription factor regulating *TGFBR1* expression, although C/EBPβ has been previously implicated in the regulation of TGF-β expression (Ma et al, 2025). To further investigate whether C/EBPβ directly regulates *TGFBR1* expression, CUT&Tag analysis was performed to identify the binding site of C/EBPβ in the *TGFBR1* promoter region. The genome browser track analysis of C/EBPβ CUT&Tag data in *TGFBR1* promoter regions demonstrated markedly increased binding signals in the binding region (−1409/−570) when compared to negative controls (Fig. 5R). To further verify this, we designed two pairs of primers on the promoter region. As expected, the enrichment of this region increased dramatically (Fig. 5S,T). Next, we generated human *TGFBR1* promoter/luciferase reporter constructs encompassing the C/EBPβ binding region on the *TGFBR1* promoter. C/EBPβ increased luciferase activities when compared to the empty vector (Fig. 5U), indicating that C/EBPβ directly regulates *TGFBR1* promoter activation. Additionally, the mutation of four main C/EBPβ binding motifs within the promoter region significantly reduced luciferase activity (Figs. 5V and EV11), providing further evidence of C/EBPβ directly regulating *TGFBR1* transcription. Taken together, these data suggest that C/EBPβ is a novel regulator of *TGFBR1* to activate TGF-β signaling.

## C/EBPβ drives the fibroblast-like EC phenotype and atherosclerosis in vivo

Our findings suggest that C/EBPβ promotes endothelial pathological phenotype transition and drives atherosclerosis at an early stage. To further evaluate this in vivo, endothelial-specific AAV-C/EBPβ was constructed and injected into mice via the tail vein prior to HFD feeding. 8-week-old *Apoe*⁻/⁻ mice with AAV-Vector or AAV-C/EBPβ were placed on a high-fat diet and then sacrificed 8 weeks later. The expression of endothelial Flag-C/EBPβ was confirmed by immunofluorescence (IF) staining (Fig. EV12A). Plasma lipid profiling revealed no significant differences in triglycerides, total cholesterol, or LDL-C between AAV-C/EBPβ and control (AAV-Vector) mice (Fig. EV12B–D). The en-face Oil Red O staining of the whole aorta was used to assess the total atherosclerotic plaque area. Compared with control mice, mice with

C/EBPβ overexpression showed a marked increase in atherosclerotic area along the length of the entire aorta (Fig. 6A,B). Histologic analysis of aortic root sections and brachiocephalic trunk sections further confirmed enhanced atherosclerosis in AAV-C/EBPβ mice (Fig. 6C,D). In agreement with the overall increase of the lesion area, immunostaining of the aortic root sections demonstrated an increase in the number of blood-derived monocytes/macrophages (CD68) and VCAM1 expression (Fig. 6E–H). Moreover, increased endothelial ICAM1, FN1 and TGFBR1 were detected in AAV-C/EBPβ mice (Fig. 6I–N), consistent with our in vitro findings. Taken together, these findings suggest that the upregulation of endothelial C/EBPβ promotes inflammation, endothelial dysfunction, and atherosclerosis.

## Discussion

The advent of single-cell transcriptomics has revolutionized our understanding of cellular heterogeneity in complex diseases such as atherosclerosis. In the present study, we identified a unique fibroblast-like EC subset that emerges during atherosclerotic progression, characterized by the high expression of fibroblast and pro-inflammatory genes with a reduction in EC fate. Moreover, the marker genes in this EC population increased with plaque development, indicating their crucial role in atherosclerosis progression. Further analysis revealed that C/EBPβ, the most upregulated transcription factor in these cells, acts as a novel regulator driving EC fibroblast-like and the inflammatory phenotype of EC in vitro by enhancing TGF-β signaling through the direct upregulation of TGFBR1 expression. Accelerated inflammation and atherosclerosis were confirmed by the endothelial-specific overexpression of C/EBPβ in vivo. To the best of our knowledge, these findings are the first to define the gene profile of the pathological fibroblast-like EC in detail and to identify endothelial C/EBPβ as a transcription factor of *TGFBR1*, which drives this fibroblast-like EC fate and aggravates atherosclerosis. These findings highlight potential therapeutic targets for mitigating vascular inflammation and atherosclerosis.

Using scRNA-seq, these atherogenic fibroblast cells were defined by the high expression of fibroblast marker genes (*Dcn*, *Fn1*, *Lum*, *Ccl2*, and *Pdgfrα*), high expression of inflammatory genes (*ICAM1*, *VCMA1*, *SELE*, *CXCL1*, *CXCL2*), and reduced expression of EC fate marker genes (Fig. 2A). These fibroblast marker genes and inflammation-related genes have been reported to play important roles in monocyte adhesion, macrophage migration, and the transformation of monocytes into macrophages, thereby leading

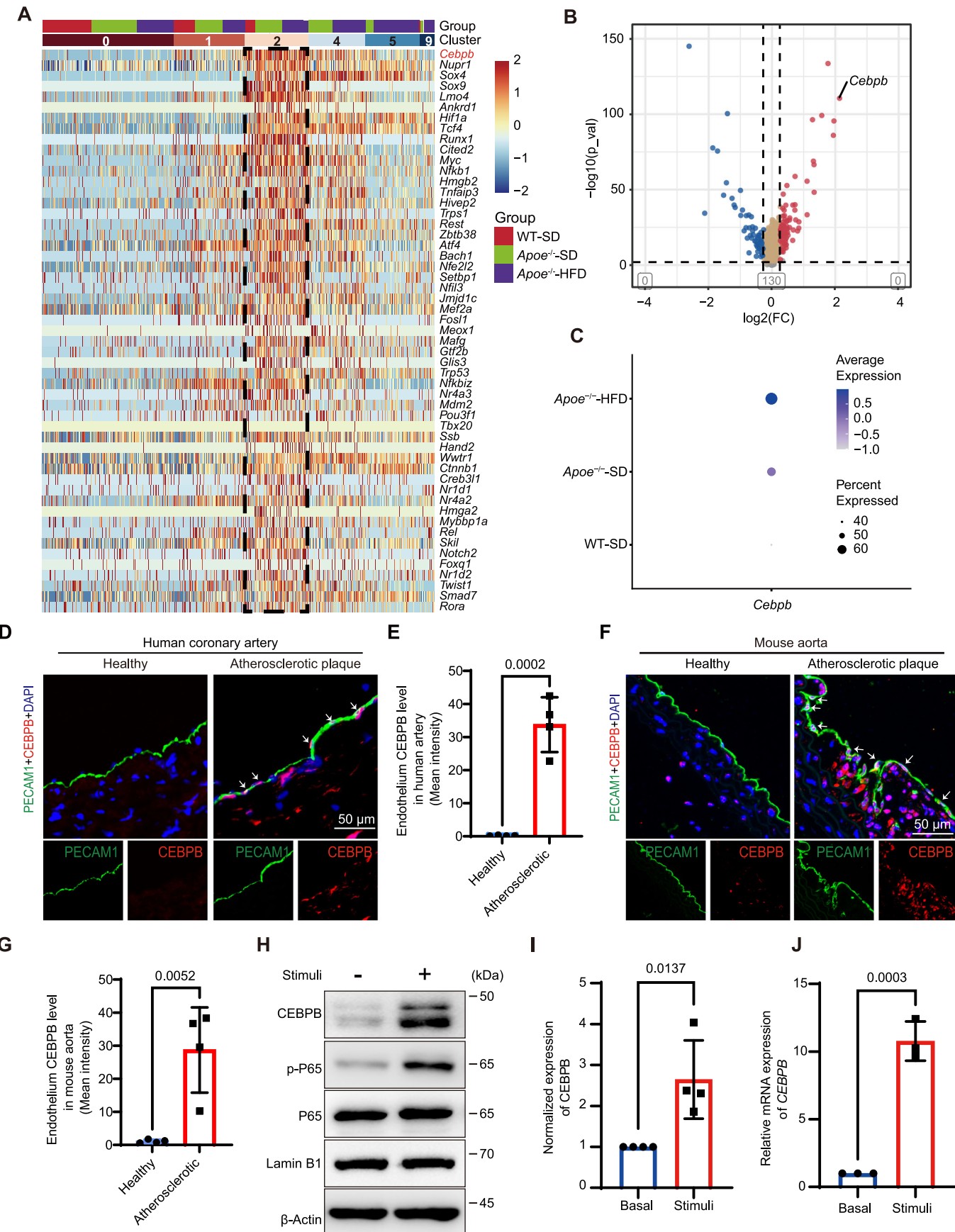

**Figure 3.   High expression of endothelial C/EBPβ is associated with atherosclerosis.**

(A) Heatmap revealing the expression of upregulated transcription factors in cluster 2 (indicated by a black dashed box) compared to clusters 0, 1, 4, 5, and 9, arranged by cluster and group. (B) Volcano plot showing that the expression of *Cebpb* is most upregulated in cluster 2 compared with cluster 0 and 1. (C) Dot plot showing average expression and proportion of *Cebpb*-expressing cells in ECs from normal aortas (WT-SD), mild atherosclerotic aortas (*Apoe*$^{-/-}$-SD), and severe atherosclerotic aortas (*Apoe*$^{-/-}$-HFD). (D) Representative IF staining of PECAM1 (green), C/EBPβ (red), and DAPI (blue) for healthy and plaque-associated endothelium in human coronary arteries. White arrows indicate PECAM1/C/EBPβ colocalization. Scale bar, 50 μm. (E) Quantification of the mean intensity (gray value) of endothelium C/EBPβ level in human artery (*n* = 4). (F) Representative IF staining of PECAM1 (green), C/EBPβ (red), and DAPI (blue) for healthy and plaque-associated endothelium in *Apoe*$^{-/-}$ mouse aortas. White arrows indicate PECAM1/C/EBPβ colocalization. Scale bar, 50 μm. (G) Quantification of mean intensity (gray value) of endothelium C/EBPβ level in mouse aortas (*n* = 4). (H, I) Western blot analysis of C/EBPβ expression in control or inflammatory cytokines (IL-1β and TNF-α)-treated HAECs (*n* = 4). Cells were treated with the vehicle or cytokines for 48 h before harvest. Each cytokine was used at 10 ng/mL. All blots were performed on separate membranes and were not subjected to reprobing. Quantification of C/EBPβ relative to Lamin B1 is shown in (I). (J) Normalized mRNA expression level of *CEBPB* in control or inflammatory cytokines-treated HAECs, as measured by RT-qPCR. Cells were treated as in (H). *n* = 3 biological replicates. Data in (E, G, I, J) are shown as mean ± SD. Normal distribution was confirmed by the Shapiro–Wilk test, and differences were analyzed using an unpaired *t*-test (two-tailed). *P* < 0.05 was considered statistically significant. Source data are available online for this figure.

to atherosclerosis progression (Georgakis et al, 2022; Hedayati-Moghadam et al, 2021; Rohwedder et al, 2012; Sagris et al, 2023; Singh et al, 2023; Talusan et al, 2005), indicating the detrimental role of these cells in atherogenesis. In addition, pseudotime trajectory analysis further supports this hypothesis, positioning fibroblast-like ECs as terminal effector cells in the EC phenotypic continuum. The decline in clusters 0 and 1—enriched in metabolic and homeostatic pathways—parallels the loss of functional endothelium observed in advanced atherosclerosis (Gimbrone and Garcia-Cardena, 2016). Conversely, the expansion of cluster 2 correlates with disease severity, suggesting that fibroblast-like ECs may serve as both biomarkers and mediators of plaque initiation (Chen et al, 2019). Nevertheless, in the severe group, the cell proportion was reduced to 9.08%. There may be two reasons for this: first, the transition of normal endothelial cells into fibroblast-like ECs occurs at the early stage of atherosclerosis, entering into a relatively stable state thereafter; for example, we counted 116 cells in the early stage and 113 cells in the late stage (Table EV3). At the late stage of the disease, the cell counts of cluster 5 ECs increase broadly, indirectly resulting in a lower proportion of cluster 2 ECs.

Intriguingly, in addition to cluster 2, we found that cluster 5 possessed some characteristics of EndMT, such as the high expression of *Col8a1*, *Fn1*, *Twist1*, *Serpine1*, *Mmp2*, and *Fbln5* (Fig. 2B). Moreover, cells in cluster 5 showed higher expression of the SMC marker *Acta2*, which is characteristic of myofibroblast-like cells (Fig. 2A,B). Violin plots of selected marker genes further illustrate the differences among these two clusters, showing that cluster 2 is enriched in fibroblast marker genes, including *S100a4*, *Lum*, *Col1a2*, *Pdgfra*, *Dcn*, *Cxcl1*, and *Cxcl2*, while cluster 5 shows high expression levels of myofibroblast marker genes, including *Tagln*, *Acta2*, *Eln*, *Fbln5*, and *Lrg1* (Fig. EV13A), which are known to maintain vascular elasticity, inhibit inflammation, and stabilize plaque, suggesting a beneficial role for atherosclerosis (Darby et al, 2016; Gole et al, 2022; Newman et al, 2018). In accordance with the clustering analysis, the gene expression pattern of clusters 2 and 5 was also found to differ in terms of the functional enrichment annotations of DEGs as assessed by comparing the GO and KEGG terms (Fig. EV13B–E). Moreover, the pseudotime trajectory analysis also showed that clusters 2 and 5 transitioned toward different positions (Figs. 2E and EV13F). These findings suggest that clusters 2 and 5 represent two different cell fates. Cluster 2 promotes disease progression, while cluster 5 increases with the development of atherosclerotic plaque, and may help to maintain

plaque stability. Nevertheless, more studies are required to explore the functional differences between cluster 2 and cluster 5 in atherosclerosis.

Our identification of C/EBPβ as a key transcriptional driver of fibroblast-like ECs adds a new layer to the understanding of endothelial regulation. Previous studies have shown that C/EBPβ is important for regulating immune response, EC proliferation, and endothelial permeability (Ni et al, 2010; Roy et al, 2002; Tang et al, 2018; Zemskov et al, 2022). However, its involvement in endothelial pathobiology, vascular inflammation and atherosclerosis has remained unclear to date. In the present study, we demonstrate that endothelial C/EBPβ is induced by pro-inflammatory cytokines (IL-1β, TNF-α) and escalates with EC dysfunction as well as atherosclerotic severity (Figs. 3, 4, 6). Moreover, it functions as a direct activator of TGF-β signaling through *TGFBR1* transcription. Therefore, C/EBPβ may act as an earlier factor to activate TGF-β signaling and drive atherosclerosis by upregulating TGFBR1 expression. This is consistent with studies showing that SMAD2/3 activation promotes EndMT and plaque instability (Chen et al, 2019). However, our work uniquely links these effects to C/EBPβ, providing a transcriptional mechanism for TGF-β hyperactivation in diseased endothelium. Furthermore, our findings align with a previous study demonstrating that C/EBPβ drives the phenotypic transition of vascular smooth muscle cells (VSMCs) under hyperlipidemia, contributing to arterial stiffness through Daam1-mediated cytoskeletal remodeling and PDGF-CC signaling (Ma et al, 2025). While our study highlights the role of C/EBPβ in endothelial dysfunction, these results collectively suggest a conserved mechanism wherein C/EBPβ acts as a central regulator across multiple vascular cell types in atherosclerosis. Notably, future research should explore whether C/EBPβ inhibition could synergistically target both ECs and VSMCs to combat arterial stiffening and atherosclerosis. Recently, SOX4 was found to be a novel phenotypic regulator of ECs in atherosclerosis (Cheng et al, 2023). SOX4 can be induced by IL-1β and TGF-β to cause a loss of endothelial markers and gain of mesenchymal markers. However, the relationship between SOX4, C/EBPβ, and TGFBR1 requires further investigation. Taken together, our findings suggest that targeting endothelial C/EBPβ may offer new therapeutic avenues for atherosclerotic diseases.

Normally, TGF-β signaling is suppressed in quiescent endothelial cells due to low TGFBR1 expression. Previous studies showed that TGFBR1 expression was regulated by FGFR1 signaling and

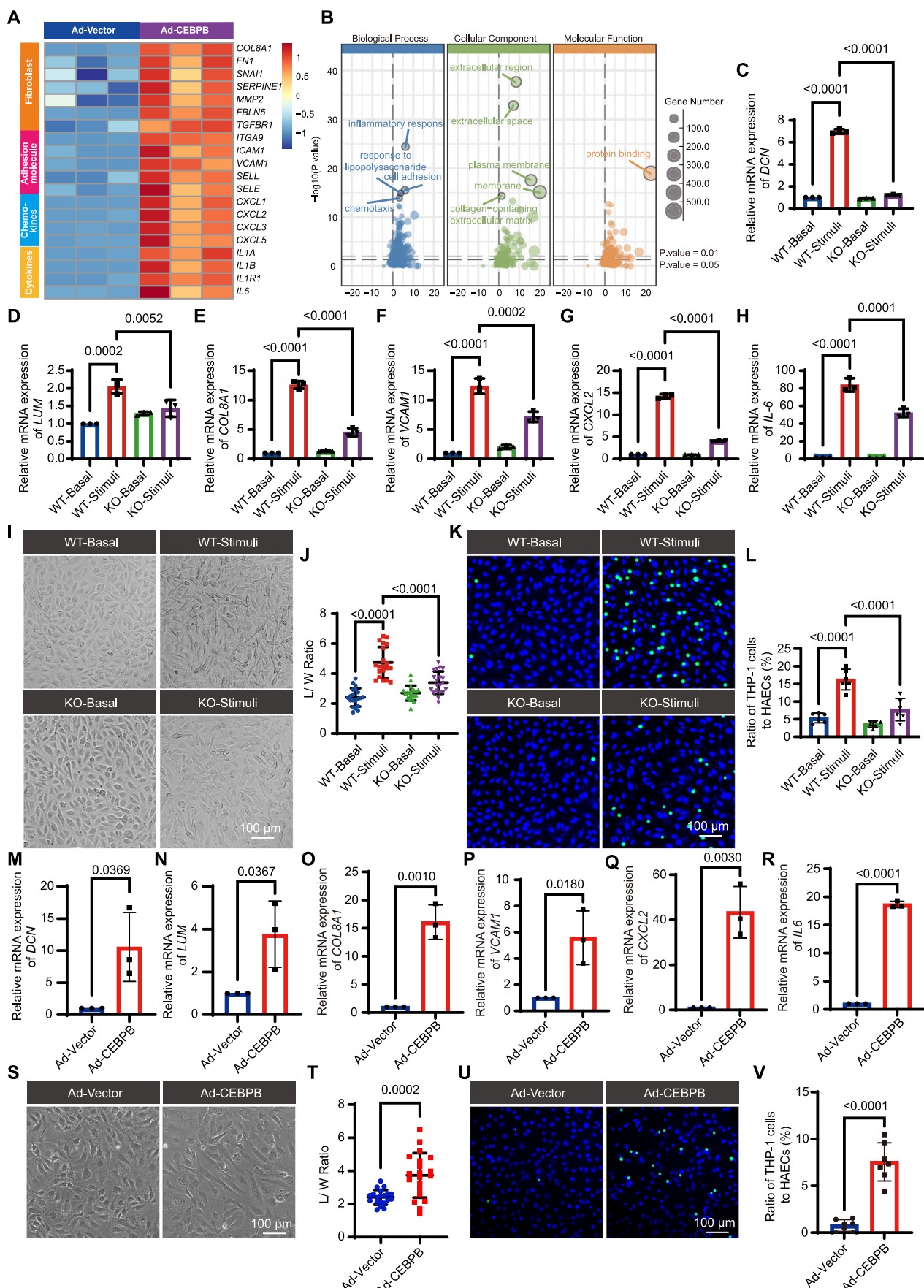

**Figure 4. C/EBPβ functions as a master regulator of the fibroblast-like EC phenotype.**

(A) Heatmap of the relative gene expression for fibroblast fate genes and inflammation-related genes in HAECs infected with Ad-Vector or Ad-C/EBPβ according to the results of RNA-seq. (B) Dot plot of GO items enriched in DEGs in HAECs infected with Ad-Vector or Ad-C/EBPβ according to the results of RNA-seq. (C–H) Relative mRNA level of *DCN, LUM, COL8A1, VCAM1, CXCL2,* and *IL6* in wildtype and C/EBPβ KO HAECs treated by the vehicle or cytokines (TNF-α + IL-1β, 10 ng/mL for each) for 48 h (n = 3). (I, J) Morphology analysis of wildtype and C/EBPβ KO HAECs treated by the vehicle or cytokines (TNF-α + IL-1β, 10 ng/mL for each) for 48 h. Representative bright field images are shown in (I) with quantification of the length-to-width (L/W) ratio of cells in (J) (n = 20). Scale bar, 100 μm. (K, L) C/EBPβ deficiency inhibits THP-1 cell adhesion to HAECs. Cells were treated as in (I). Representative images with adherent THP-1 cells (green) are shown in (K), with the relative quantification of the ratio of adherent THP-1 cells to HAECs shown in (L) (n = 6). Scale bar, 100 μm. (M–R) Relative mRNA level of *DCN, LUM, COL8A1, VCAM1, CXCL2,* and *IL6* in HAECs infected with Ad-Vector or Ad-C/EBPβ (n = 3). (S, T) Morphological analysis of HAECs infected with Ad-Vector or Ad-CEBPB. Representative bright field images are shown in (S) with quantification of the length-to-width (L/W) ratio of cells in (T) (n = 20). Scale bar, 100 μm. (U, V) Upregulation of C/EBPβ promotes THP-1 cell adhesion to HAECs. Representative images with adherent THP-1 cells (green) are shown in (U), with the relative quantification of the ratio of adherent THP-1 cells to HAECs shown in (V) (n = 7). Scale bar, 100 μm. Values in (C–H, J, L, M–R, T, V) are shown as mean ± SD. The normal distribution of (C–H) and (M–R) was confirmed by the Shapiro–Wilk test. Differences were analyzed by one-way ANOVA followed by Tukey's test or unpaired *t*-test (two-tailed). *P* < 0.05 was considered statistically significant. Source data are available online for this figure.

TGFBR1 acetylation (Chen et al, 2012; Zhu et al, 2023). Upon pro-inflammatory stimulation, FGF2-FGFR1 signaling reduces let-7 miRNA levels and enhances TGFBR1 RNA stability, thereby increasing TGFBR1 expression and driving EndMT. As soon as the EndMT process begins, EndMT induces an Ac-CoA production increase, thereby leading to TGFBR1 acetylation, which results in the activation and long-term stabilization of TGF-β signaling. Nevertheless, how TGFBR1 is transcriptionally regulated remains unknown. In the present study, we determined that C/EBPβ is a novel regulator that directly binds to the promoter of *TGFBR1* and regulates TGFBR1 expression. In addition, C/EBPβ can be induced by IL-1β and TNF-α stimulation. Based on these findings, we propose a novel mechanism for the activation of TGF-β signaling: In the early stage of atherosclerosis, elevated pro-inflammatory cytokines in the vascular inflammatory environment stimulate the expression of endothelium C/EBPβ, which then increases TGFBR1 transcription. After transcription, decreased let-7 miRNA induced by FGFR1 signaling stabilizes TGFBR1 RNA and further increases the expression of TGFBR1, which promotes the initiation of EndMT. With EndMT progression, increased Ac-CoA production enhances TGFBR1 acetylation and improves TGFBR1 protein stability, leading to the long-term activation of TGF-β signaling and resulting in the atherogenic phenotype of EC.

Despite its value, there are several limitations to our study. The first relates to the number of cells used for sequencing. From three groups of mice, we captured 446, 755, and 1302 ECs, respectively. Although this is sufficient to support the findings of this study, more cells may provide further insights into rare EC clusters. Second, we used *Apoe*$^{-/-}$ mice fed on normal or high-fat diets for a long time to simulate the different degrees of atherosclerosis, since atherosclerosis is a chronic illness. However, the future sequencing of ECs taken from *Apoe*$^{-/-}$ mice fed a high-fat diet at a different time point may provide new information. Third, further scRNA-seq of ECs from the aortas of female *Apoe*$^{-/-}$ mice would help to exclude the effect of gender on endothelial cell plasticity in atherosclerosis.

In summary, our work unveils a previously unrecognized fibroblast-like EC subset that drives atherosclerotic progression through the C/EBPβ-dependent activation of TGF-β signaling. By linking endothelial inflammation, ECM remodeling, and monocyte recruitment, this axis represents a promising target for precision medicine in atherosclerosis. Therefore, we believe that these findings may help elucidate the molecular mechanism of endothelial inflammation and atherosclerosis progression, providing a potential target for atherosclerosis treatment.

## Methods

### Reagents and tools table

| Reagent/resource | Reference or source | Identifier or Catalog Number |
|---|---|---|
| **Experimental models** | | |
| C57BL/6 (*M. musculus*) | GemPharmatech | N000013 |
| Apoe$^{-/-}$ mice (*M. musculus*) | GemPharmatech | T001458 |
| HAECs (*H. sapiens*) | Lonza | CC-2535 |
| THP-1 (*H. sapiens*) | National Collection of Authenticated Cell Cultures | SCSP-567 |
| HEK 293 T/17 (*H. sapiens*) | ATCC | CRL-11268 |
| **Recombinant DNA** | | |
| lentiCRISPRv2 | Addgene | 52961 |
| lentiCRISPRv2-CEBPB #1 | This study | N/A |
| lentiCRISPRv2-CEBPB #2 | This study | N/A |
| lentiCRISPRv2-CEBPB #3 | This study | N/A |
| pMD2.G | Addgene | 12259 |
| psPAX2 | Addgene | 12260 |
| pEnter-CEBPB | This study | N/A |
| pGL4.10-TGFBR1 promoter wt | This study | N/A |
| pGL4.10-TGFBR1 promoter mutants | This study | N/A |
| pRL-TK | Promega | E2241 |
| **Antibodies** | | |
| CD31 | Invitrogen | 11-0311-82 |
| CD45 | Invitrogen | 12-0451-82 |
| Flag-tag | Cell Signaling Technology | 14793 |
| CEBPB | Abcam | ab32358 |
| p-p65 | Cell Signaling Technology | 3033 |

| Reagent/resource | Reference or source | Identifier or Catalog Number |
|---|---|---|
| P65 | Cell Signaling Technology | 8242 |
| β-Actin | Abclonal | AC026 |
| TGFBR1 | Abcam | ab235578 |
| p-SMAD2 | Cell Signaling Technology | 18338 |
| SMAD2 | Cell Signaling Technology | 3122 |
| HRP-linked anti-Rabbit IgG secondary antibody | Cell Signaling Technology | 7074 |
| PECAM1 | R&D | AF3628 |
| CD68 | BioRad | MCA1957GA |
| VCAM1 | BD | 553330 |
| ICAM1 | BioLegend | 116102 |
| FN1 | Proteintech | 15613-1-AP |
| TGFBR1 | Proteintech | 30117-1-AP |
| LUM | Abcam | ab168348 |
| Alexa Fluor 488-conjugated Donkey anti-Goat IgG | Invitrogen | A11055 |
| Alexa Fluor 488-conjugated Donkey anti-Rat IgG | Invitrogen | A21208 |
| Alexa Fluor 594-conjugated Donkey anti-Rabbit IgG | Invitrogen | A21207 |
| Alexa Fluor 594-conjugated Donkey anti-Goat IgG | Invitrogen | A11058 |
| Mouse IgG | Abmart | B30010 |
| Flag-tag | Abmart | M20008 |
| Lamin B1 | Abclonal | A11495 |
| **Oligonucleotides and other sequence-based reagents** | | |
| qPCR primers | This study | Table EV5 |
| CEBPB sg1-Forward: CACCGCGCTTACCTCGGCTACCAGG | This study | N/A |
| CEBPB sg1-Reverse: AAACCCTGGTAGCCGAGGTAAGCGC | This study | N/A |
| CEBPB sg2- Forward: CACCGAGTACGGCTACGTGAGCCTG | This study | N/A |
| CEBPB sg2- Reverse: AAACCAGGCTCACGTAGCCGTACTC | This study | N/A |
| CEBPB sg3- Forward: CACCGTCGACTTCAGCCCGTACCTG | This study | N/A |
| CEBPB sg3- Reverse: AAACCAGGTACGGGCTGAAGTCGAC | This study | N/A |
| CEBPB siRNA- sense: AGCACAGCGACGAGUACAAGA | This study | N/A |
| CEBPB siRNA- anti-sense: UUGUACUCGUCGCUGUGCUG | This study | N/A |
| TGFBR1 siRNA- sense: CCAUCGAGUGCCAAAUGAAUU | This study | N/A |
| TGFBR1 siRNA- anti-sense: UUCAUUUGGCACUCGAUGGTT | This study | N/A |
| **Chemicals, Enzymes and other reagents** | | |

| Reagent/resource | Reference or source | Identifier or Catalog Number |
|---|---|---|
| Isoflurane | RWD | R510-22-10 |
| Collagenase A | Roche | 10103578001 |
| Elastase | Sigma-Aldrich | E1250 |
| DMEM | Gibco | 11995065 |
| ECM | ScienCell | 1001 |
| RPMI 1640 | Gibco | 11875093 |
| Opti-MEM | Gibco | 31985088 |
| FBS | Noverse | NFBS2500A |
| Penicillin/streptomycin | Gibco | 15140122 |
| DAPI | Solarbio | C0060 |
| Protease Inhibitor cocktail | LABLEAD | C0101 |
| Phosphatase inhibitors cocktail | LABLEAD | C0104 |
| OCT | Sakura | 4583 |
| Oil Red O | Sigma-Aldrich | O0625 |
| Modified Oil Red O Staining Kit | Beyotime | C0158M |
| Lentivirus Concentration Reagent | Life-ilab | AC04L442 |
| Puromycin | MCE | HY-K1057 |
| Calcein-AM | YEASEN | 40719ES50 |
| Hoechst 33342 | Invitrogen | H3570 |
| FITC-dextran | Sigma-Aldrich | FD40S |
| Hieff Trans liposomal transfection reagent | YEASEN | 40802ES03 |
| Hieff Trans siRNA/miRNA reagent | YEASEN | 40806ES03 |
| Trizol reagent | Ambion | 15596026 |
| SteadyPure Universal RNA Extraction Kit II | Accurate Biology | AG21022 |
| mRNA reverse kit | Accurate Biology | AG11706 |
| SYBR master mix | Accurate Biology | AG11701 |
| Enhanced ECL Chemiluminescent Substrate Kit | YEASEN | 36222ES60 |
| Hyperactive Universal CUT&Tag Assay Kit for Illumina kit | Vazyme Biotech | TD904 |
| Dual Luciferase Reporter Gene Assay Kit | YEASEN | 11402ES60 |
| **Software** | | |
| GraphPad Prism 9 | https://www.graphpad.com | |
| ImageJ | https://imagej.net/ij/ | |
| Olyvia | https://olyvia.software.informer.com/ | |
| OmicStucio tools | https://www.omicstudio.cn/tool | |
| OmicShare | https://www.omicshare.com/tools | |
| Cut_tag_tool | http://cloud.vazyme.com:83/ | |

| Reagent/resource | Reference or source | Identifier or Catalog Number |
|---|---|---|
| OmicStudio | https://www.omicstudio.cn/cell | |
| OmicStucio tools | https://www.omicstudio.cn/tool | |
| Other | | |
| BD FACS ORP ARIA II | BD biosciences | |
| Illumina Novaseq 6000 | Illumina | |
| SLIDEVIEW VS200 | Olympus | |
| FV3000 | Olympus | |
| FlexStation 3 | Molecular Devices | |
| VICTOR Nivo | PerkinElmer | |

## Animals and diet

8-weeks-old male C57BL/6 and Apoe$^{-/-}$ mice (in a C57BL/6 background) were purchased from GemPharmatech Co., Ltd. Four C57BL/6 mice and four Apoe$^{-/-}$ mice were fed a standard diet, while another four Apoe$^{-/-}$ mice were fed a high-fat diet (40% kcal fat, 1.25% cholesterol, 0% cholic acid, D12108C, SYSE Bio-tech. Co., Ltd) for 14 weeks. For EC-specific CEBPB overexpression, Apoe$^{-/-}$ mice were injected with $1 \times 10^{12}$ VG/mL AAV-Vector or AAV-CEBPB (AAV-Tie-Cebpb-flag, Genomeditech, Shanghai, China) through the tail vein, followed by a high-fat diet for 8 weeks. All mice were kept in the laboratory animal center of Zhejiang University under a 12 h/12 h light/dark cycle and in a pathogen-free environment. All samples were collected at a single endpoint to avoid age-related pathological changes. All mice were double-blinded and randomly selected to minimize the effects of subjective bias. All research followed the Guidelines on the Care and Use of Laboratory Animals (National Institutes of Health Publication no. 85-23, revised 1996), and animal study protocols were approved by the institutional Animal Care and Use Committee of Zhejiang University (Approved code: ZJU20230183).

## Preparation of a single-cell EC suspension

All animals were anaesthetized by isoflurane inhalation (3%) and euthanized by exsanguination (RWD, R510-22-10). The adequacy of anesthesia was confirmed by the lack of a toe pinch withdrawal response during the surgical procedure according to the Animal Euthanasia Guidelines of Zhejiang University, and each animal was immediately perfused with 10 mL of saline through the left ventricle. Aortas from four mice in each group were harvested and dissected into small pieces. Tissues were digested in a 1.5 mL digestive solution containing 1.5 mg/mL Collagenase A (Roche, 10103578001) and 0.5 mg/mL Elastase (Sigma-Aldrich, E1250) dissolved in DMEM (Gibco) with 10% FBS (Noverse, NFBS2500A) at 37 °C for 3 h. Suspension was slowly pipetted up and down using sterile pipette tips every 30 min. After digestion, the solution was transferred into flow tubes, and 3 mL of flow buffer (i.e., PBS with 10% FBS) was added to terminate the digestion. After

centrifugating at 500×g for 5 min at 4 °C, the supernatant was discarded, and the pellet was washed with 1 mL of flow buffer followed by centrifugation as described above. Cell pellets were then resuspended in 150 μL of flow buffer and incubated with fluorochrome-coupled antibodies against CD31 (Invitrogen, 11-0311-82) and CD45 (Invitrogen, 12-0451-82) at 1:100 for 30 min on ice in the dark. After incubation, cells were washed with 1 mL of flow buffer and resuspended in 300 μL of PBS. 4,6-Diamidino-2-phenylindole (DAPI) (1:1000) was added to mark dead cells before sorting. The resulting cell suspension was filtered using 40-μm cell strainers. The living CD45$^-$ and CD31$^+$ cells were then acquired using a flow sorter (BD FACS ORP ARIA II).

## Single-cell RNA sequencing (scRNA-seq)

Cells obtained by flow sorting were separated by microfluidic technology using a 10× Genomics platform. Next, single cells, barcoded gel beads, and primers were coated with oil droplets containing enzymes to form gel beads in emulsions (GEMs). The mRNAs released from cells were then reverse transcribed into cDNA fragments and labeled with a barcode. The resulting cDNAs were then used as a template for PCR amplification to construct a standard sequencing library. An Illumina Novaseq 6000 sequencing platform (LC-Bio Technologies (Hangzhou) Co., Ltd) was used for high-throughput sequencing.

## scRNA-seq data processing and analysis

Cell Ranger (V4.0.0) was used for the quality control and alignment of data to the reference genome (i.e., Mus_musculus.GRCm38.96). Data from three groups were combined after normalizing the read depth. Seurat (R package, V3.1.1) was used to filter low-quality cells and obtain high-quality cells (the number of genes identified in a single cell was more than 500 and the proportion of mitochondrial gene expression in a single cell was no more than 25%).

Normalization and Principal Component Analysis (PCA) were performed by Seurat. The top 20 principal components were selected from the PCA, and cells were then clustered with a resolution of 0.8. T-distributed neighbor embedding (t-SNE) was used to visualize this analysis. The FeaturePlot, VlnPlot, DotPlot, DoHeatmap, and DimPlot functions of Seurat were used to generate gene expression t-SNE plots, violin plots, dot plots, gene expression heatmaps, and highlight cluster t-SNE plots. Marker genes for each cluster were determined using Bimod (i.e., using a likelihood-ratio test) via the FindAllMarkers function implemented in Seurat (P value ≤0.01, Log2 fold change ≥0.26, percentage of cells where a gene was detected in a specific cluster >10%). To identify biologically relevant cell clusters for trajectory reconstruction, we first generated a cluster dendrogram and calculated Pearson correlation coefficients between the average gene expression profiles of all ten clusters. Both steps were performed using OmicStudio tools (https://www.omicstudio.cn/tool). Based on the dendrogram topology and Pearson correlation matrix, clusters 0, 1, 2, 4, 5, and 9 were identified as closely related to cluster 2. Therefore, these clusters were selected for downstream trajectory analysis. ECs from the six selected clusters were subjected to pseudotime analysis using Monocle 2 (https://www.omicstudio.cn/cell). GO analysis of DEGs was performed using the OmicStucio tools (https://www.omicstudio.cn/tool). Cluster dendrogram and

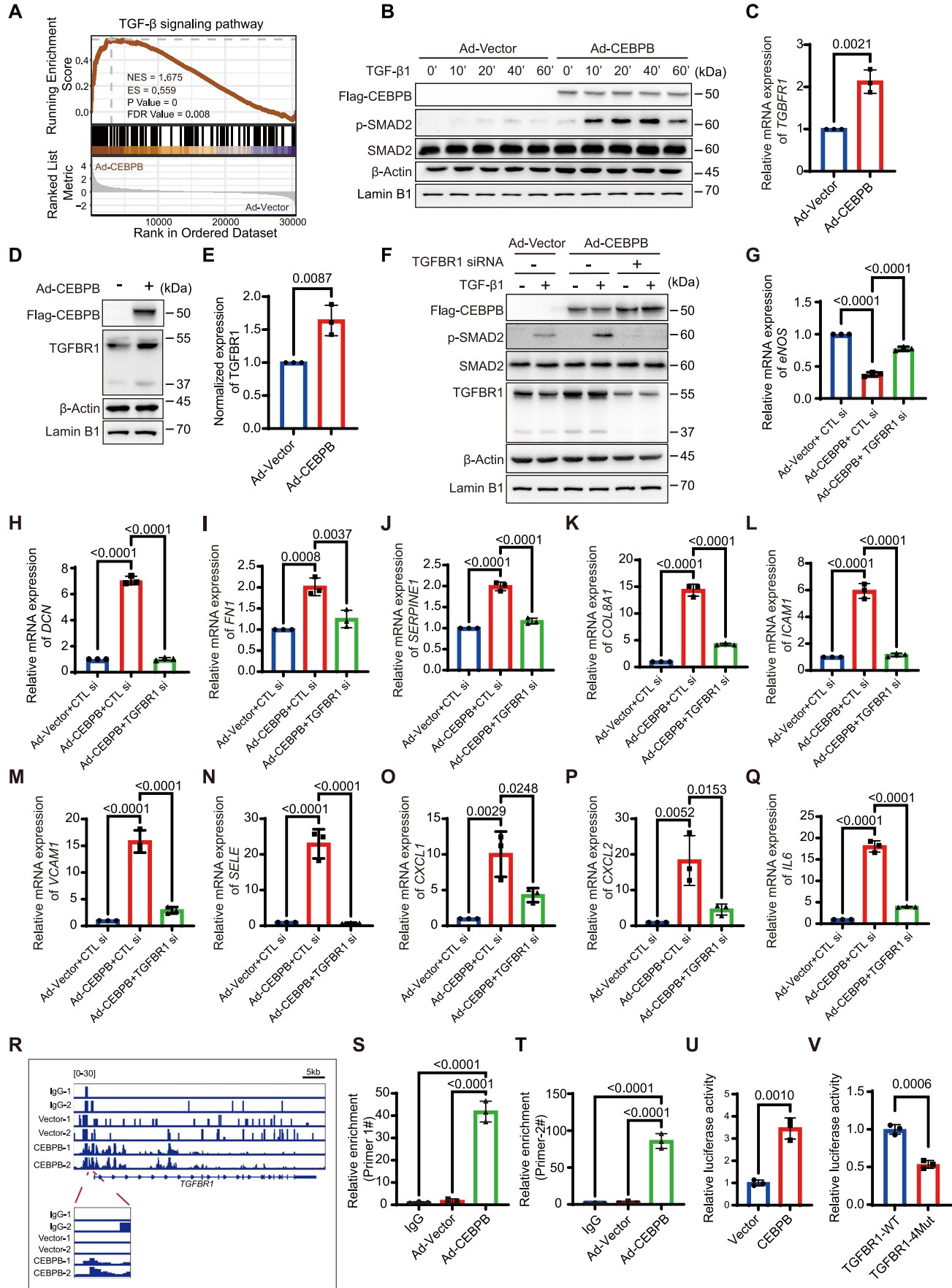

**Figure 5.  C/EBPβ activates TGF-β signaling by directly upregulating TGFBR1 expression.**

(A) GSEA plots showing the increased expression of gene sets involved in the TGF-β signaling pathway according to the RNA-seq results. NES, ES, P-value and FDR value (FDR-corrected *P* value) are presented. (B) Representative blots of p-SMAD2 and SMAD2 in HAECs. Cells were infected with Ad-Vector or Ad-C/EBPβ followed by TGF-β1 treatment according to various time points as indicated ($n = 3$). (C) Normalized mRNA expression of *TGFBR1* in HAECs infected with Ad-Vector or Ad-C/EBPβ ($n = 3$). (D, E) Western blot analysis of TGFBR1 in HAECs after Ad-C/EBPβ infection. Representative blots are shown in (D), with relative quantification in (E) ($n = 3$). (F) Representative blots of p-SMAD2, SMAD2, and TGFBR1 in HAECs. Cells were transfected with negative control siRNA or TGFBR1 siRNA for 24 h followed by Ad-Vector or Ad-C/EBPβ infection for another 72 h and then treated with TGF-β1 for 40 min before harvest ($n = 3$). All blots in (B, D, F) were performed on separate membranes and were not subjected to reprobing. (G–Q) Relative mRNA expression of *eNOS, DCN, FN1, SERPINE1, COL8A1, ICAM1, VCAM1, SELE, CXCL1, CXCL2*, and *IL6* in HAECs ($n = 3$). Cells were transfected with negative control siRNA or TGFBR1 siRNA for 24 h, followed by Ad-Vector or Ad-C/EBPβ infection for 72 h before harvest. (R) Genome browser view of Ad-C/EBPβ Cut&Tag with two negative controls (IgG and Ad-Vector) at the *TGFBR1* promoter in HAECs. Two independent CUT&Tag experiments were sequenced. (S, T) Relative fold enrichment of the promoter region ($-1409/-570$) of *TGFBR1* ($n = 3$). (U) Dual luciferase assay indicating the transcription activity of C/EBPβ to *TGFBR1* in 293T cells ($n = 3$). Relative luciferase activity was normalized to that of Renilla and the Vector group. (V) Dual luciferase assay indicating the binding sites of C/EBPβ for *TGFBR1* transcription in 293T cells ($n = 3$). Relative luciferase activity was normalized to that of Renilla and the WT group. Values are shown as mean ± SD. Normal distribution was confirmed by the Shapiro–Wilk test. All the differences were analyzed by unpaired *t*-test (two-tailed) (C, E, U, V) or one-way ANOVA followed by Tukey's test (G–Q, S, T). $P < 0.05$ was considered statistically significant. Source data are available online for this figure.

Pearson correlation analysis were performed using OmicShare (https://www.omicshare.com/tools).

## Quantitative real-time PCR

The mRNA levels of *CEBPB, eNOS, LUM, DCN, COL8A1, SERPINE1, FN1, IL6, ICAM1, VCAM1, SELE, CXCL1, CXCL2*, and *TGFBR1* in HAECs were quantified using quantitative real-time PCR (qRT-PCR). Total cellular RNA was isolated using a SteadyPure Universal RNA Extraction Kit II (Accurate Biology, AG21022) with all procedures performed according to the manufacturer's protocol. An mRNA reverse kit (Accurate Biology, AG11706) was used to produce cDNA before the quantification of mRNA levels. For RT-PCR, a SYBR master mix (Accurate Biology, AG11701) and BioRad iCycler real-time PCR system (BioRad) were used. qPCR primers used for this study can be found in Table EV5.

## Western blotting

After treatments, the cells were washed twice with ice-cold PBS and prepared in 150 μL lysis buffer containing 150 mM NaCl, 50 mM Tris-HCl (pH 7.5), 1 mM EDTA, 1 mM EGTA, 0.6% Triton X-100, Protease Inhibitor cocktail (LABLEAD, C0101), and phosphatase inhibitors cocktail (LABLEAD, C0104). Proteins were separated by SDS-PAGE and blotted. The PVDF membranes were incubated with primary antibodies against Flag-tag (Cell Signaling Technology, 14793, 1:1000), CEBPB (Abcam, ab32358, 1:1000), p-p65 (Cell Signaling Technology, 3033, 1:500), p65 (Cell Signaling Technology, 8242, 1:1000), β-Actin (Abclonal, AC026, 1:10,000), Lamin B1 (Abclonal, A11495, 1:1000), TGFBR1 (Abcam, ab235578, 1:1000), p-SMAD2 (Cell Signaling Technology, 18338, 1:1000), SMAD2 (Cell Signaling Technology, 3122, 1:1000) overnight at 4 °C and then incubated with HRP-linked anti-Rabbit IgG secondary antibody (Cell Signaling Technology, 7074, 1:7500) for 1 h at room temperature. The blots were developed using an ECL Chemiluminescent Substrate Kit (YEASEN, 36222ES60). The protein expression levels were quantified using ImageJ and normalized to the control group.

## Clinical specimens

Human coronary arteries were collected from the explanted hearts of transplant recipients or cadaver organ donors with informed consent. The left main coronary arteries were removed within the operating room, and vascular samples were embedded in Optimal Cutting Temperature compound (OCT) (Sakura, 4583) and frozen immediately. This study conformed to the principles outlined in the Declaration of Helsinki, and the procedures were approved by the Institutional Review Boards of Yale University and the New England Organ Bank. A waiver for consent was approved for surgical patients, and written informed consent was obtained from a family member for deceased organ donors.

## Histology and IF analysis of atherosclerotic lesions

Mice were anaesthetized and perfused through the left ventricle with heparin saline solution for 5 min. After dissection, the brachiocephalic artery and heart were isolated and fixed with 4% paraformaldehyde, followed by OCT embedding. The whole aorta, including the ascending arch, thoracic segment, and abdominal segment, was dissected and gently cleaned of adventitial tissue, followed by Oil Red O (O0625, Sigma-Aldrich) staining to measure lesions in the aorta. The surface lesion area was quantified using ImageJ.

The OCT sections were dried at room temperature for 30 min before staining. Oil Red O staining was performed using a modified Oil Red O Staining Kit (Beyotime, C0158M), according to the manufacturer's protocol.

For IF staining, OCT sections were dried at room temperature for 30 min and washed in PBS to remove the OCT compound. The sections were then blocked and permeabilized by PBS solution containing 5% donkey serum, 1% BSA and 0.3% Triton X-100 for 1 h at room temperature. Next, sections were incubated with primary antibodies at 4 °C overnight. The next day, the sections were washed with PBS three times, followed by incubating with fluorescence-coupled secondary antibodies and DAPI (1:1000) at room temperature for 1 h. After another PBS wash, sections were mounted and imaged using a microscope slide scanner (SLIDE-VIEW VS200, Olympus) or confocal laser scanning microscope (FV3000, Olympus) and analyzed using Olyvia. The primary antibodies used included PECAM1 (R&D, AF3628, 1:200), CEBPB (Abcam, ab32358, 1:100), CD68 (BioRad, MCA1957GA, 1:500, 1:100), VCAM1 (BD, 553330, 1:100), ICAM1 (BioLegend, 116,102, 1:50), FN1 (Proteintech, 15613-1-AP, 1:100), TGFBR1 (Proteintech, 30117-1-AP, 1:100), LUM (Abcam, ab168348, 1:250), and Flag-tag (CST, 14793, 1:1000). The fluorescence-coupled secondary

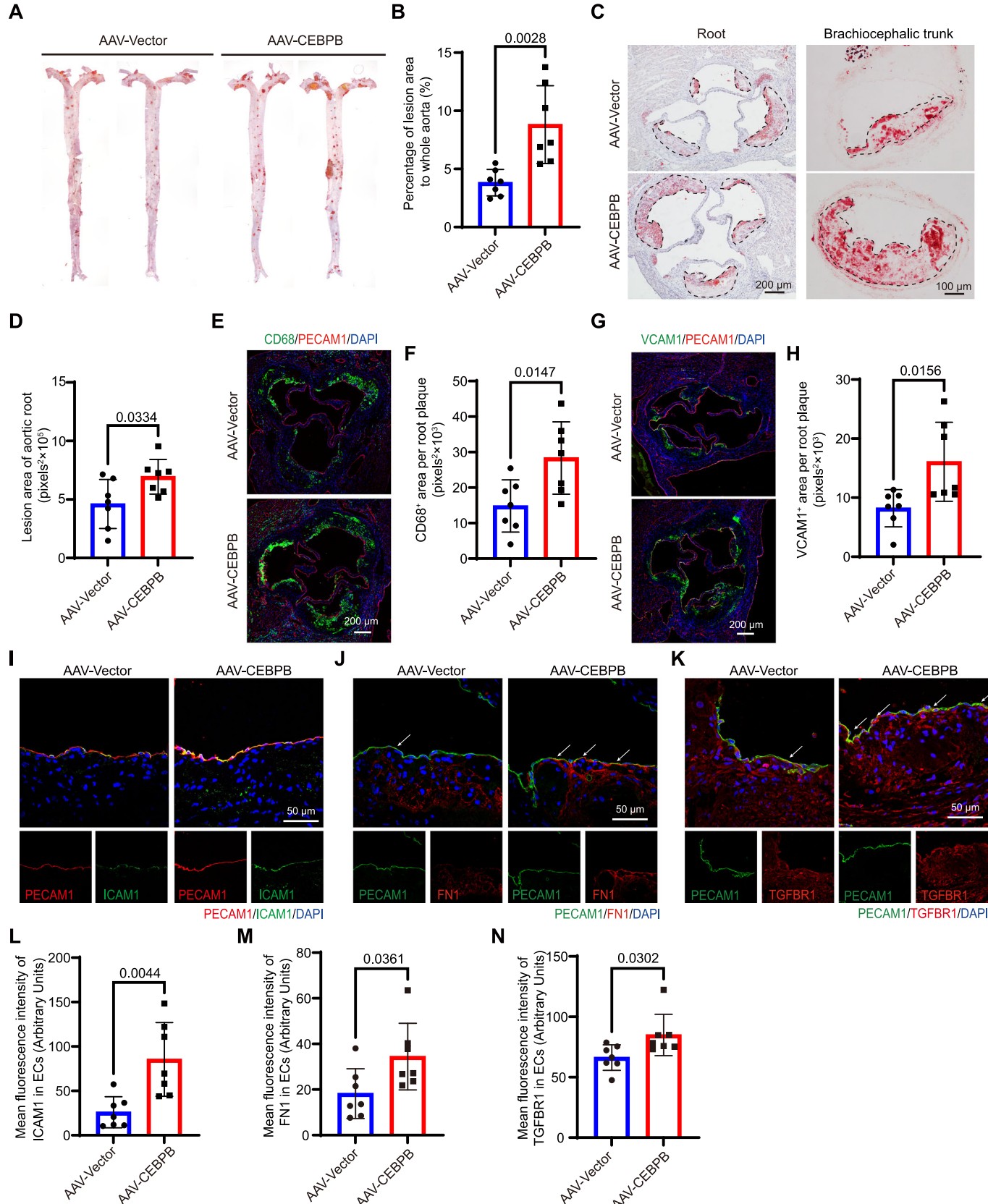

**Figure 6. Endothelial C/EBPβ exacerbates atherosclerosis in vivo.**

(A, B) Histological analysis of plaque in the whole aorta. Representative Oil Red O images of the whole aorta from AAV-Vector or AAV-C/EBPβ mice fed a HFD for 8 weeks are shown in (A), with quantification of the percentage of lesion area in whole aortas shown in (B) ($n = 7$). (C, D) Histological analysis of plaque in the aortic root and brachiocephalic trunk. Representative images of the Oil Red O staining of aortic roots (left column) and the brachiocephalic trunk (right column) from AAV-Vector or AAV-C/EBPβ mice fed a HFD for 8 weeks are shown in (B), with the quantification of plaque area in the aortic root based on Oil Red O staining shown in (D) ($n = 7$). (E, F) Representative IF staining of PECAM1 (red), CD68 (green), and DAPI (blue) from AAV-Vector or AAV-C/EBPβ mice fed with HFD for 8 weeks is shown in (E), with quantification of the CD68$^+$ area in the aortic root shown in (F) ($n = 7$). Scale bar, 200 μm. (G, H) Representative IF staining of PECAM1 (red), VCAM1 (green), and DAPI (blue) from AAV-Vector or AAV-C/EBPβ mice fed an HFD for 8 weeks is shown in (G), with quantification of the VCAM1$^+$ area in the aortic root shown in (H) ($n = 7$). Scale bar, 200 μm. (I, L) Representative IF staining of PECAM1 (red), ICAM1 (green), and DAPI (blue) from AAV-Vector or AAV-C/EBPβ mice fed a HFD for 8 weeks is shown in (I), with quantification of the mean fluorescence intensity of ICAM1$^+$ in the endothelium shown in (L) ($n = 7$). Scale bar, 50 μm. (J, M) Representative IF staining of PECAM1 (green), FN1 (red), and DAPI (blue) from AAV-Vector or AAV-C/EBPβ mice fed a HFD for 8 weeks is shown in (J), with quantification of the mean fluorescence intensity of FN1$^+$ in the endothelium shown in (M) ($n = 7$). Scale bar, 50 μm. (K, N) Representative IF staining of PECAM1 (green), TGFBR1 (red), and DAPI (blue) from AAV-Vector or AAV-C/EBPβ mice fed a HFD for 8 weeks is shown in (K), with quantification of the TGFBR1$^+$ in the endothelium shown in (N) ($n = 7$). Scale bar, 50 μm. All values are presented as mean ± SD. Statistical significance was determined using an unpaired $t$-test (two-tailed). $P < 0.05$ was considered statistically significant. Source data are available online for this figure.

antibodies used included Alexa Fluor 488-conjugated Donkey anti-Goat IgG (Invitrogen, A11055, 1:500), Alexa Fluor 488-conjugated Donkey anti-Rat IgG (Invitrogen, A21208, 1:500), Alexa Fluor 594-conjugated Donkey anti-Rabbit IgG (Invitrogen, A21207, 1:500) and Alexa Fluor 594-conjugated Donkey anti-Goat IgG (Invitrogen, A11058, 1:500). All antibodies were diluted in the solution used for blocking and permeabilization.

## Cell culture

HAECs were obtained from Lonza and maintained in ECM (ScienCell, 1001) at 37 °C in 5% $CO_2$. THP-1 cells were obtained from the National Collection of Authenticated Cell Cultures and cultured in 1640 with 10% FBS (Biochannel, BC-SE-FBS01) and penicillin/streptomycin (Gibco, 15140122) at 37 °C in 5% $CO_2$.

## CRISPR/Cas9-mediated CEBPB KO in HAECs via lentiviral transduction

The CEBPB-knockout HAECs were generated using CRISPR/Cas9. Guide RNA (gRNA) sequences targeting CEBPB were designed using the Broad Institute online tool and cloned into the lentiCRISPRv2 vector (Addgene #52961). For lentivirus production, the lentiCRISPRv2 plasmid carrying the gRNA was co-transfected with the packaging plasmids pMD2.G (Addgene #12259) and psPAX2 (Addgene #12260) into HEK 293 T cells. The virus-containing supernatant was harvested at 48 and 72 h post-transfection, filtered through a 0.45 μm low-protein-binding membrane, and concentrated using a Lentivirus Concentration Reagent (Life-ilab, AC04L442). The viral pellet was then resuspended in PBS. HAECs were seeded in six-well plates at a density of $2 \times 10^5$ cells per well 24 h before transduction. The following day, the cells were transduced with lentivirus in fresh medium containing 8 μg/mL polybrene. After 24 h, the medium was replaced. At 48 h post-transduction, puromycin selection was initiated at a concentration of 1 μg/mL (MCE, HY-K1057) and maintained until all non-transduced control cells had died, ensuring a pure population of stably transduced cells. The KO efficiency was validated by Western blot analysis. The single guide RNA (sgRNA) sequences targeting C/EBPβ tested are as follows:

CEBPB sg1-Forward: CACCGCGCTTACCTCGGCTACCAGG
CEBPB sg1-Reverse: AAACCCTGGTAGCCGAGGTAAGCGC
CEBPB sg2- Forward: CACCGAGTACGGCTACGTGAGCCTG

CEBPB sg2- Reverse: AAACCAGGCTCACGTAGCCGTACTC
CEBPB sg3- Forward: CACCGTCGACTTCAGCCCGTACCTG
CEBPB sg3- Reverse: AAACCAGGTACGGGCTGAAGTCGAC

## Adenovirus infection

Adenovirus overexpressing C/EBPβ and a mock plasmid were purchased from WZ Biosciences Inc. (Shandong, China). HAECs were allowed to grow to 70–80% confluence, followed by adenovirus infection in Opti-MEM (Gibco, 31985088) for 6 h and cultured in fresh full medium for another 72 h before conducting further experiments.

## Length-to-width ratio analysis

Cells were infected by Ad-C/EBPβ or vector control as previously described. After 72 h of infection, cells were fixed with 4% PFA for 30 min at room temperature. After washing with PBS, cells were imaged in bright field with five random fields. Four cells from each field were randomly selected for length and width measurements. The length and width measurements of cells were performed in ImageJ.

## Monocyte adhesion assay

HAECs were cultured in 12-well plates. After treatment, 2 μM Calcein-AM (YEASEN, 40719ES50) labeled THP-1 cells ($1 \times 10^5$ cells /well) were planted in HAECs and co-cultured for 1 h at 37 °C in 5% $CO_2$. After co-culture, cells were stained with Hoechst 33342 (Invitrogen, H3570) for 10 min at 37 °C in 5% $CO_2$. Cells were then washed with PBS three times to remove unadhered THP-1 cells. Each field were randomly imaged. The THP-1 cells that adhered to HAECs were counted.

## Transwell permeability assay

HAECs were seeded into a Transwell (0.4-μm, LabSelect, 14,211) at a density of $1 \times 10^5$ and cultured for 48 h (until cells were confluent). The medium in the upper and lower chambers was then discarded. Then, 200 μL of FITC-dextran (1 mg/mL, average mol. wt. 40,000, FD40S, Sigma-Aldrich) was added to the upper chamber. Thereafter, 500 μL of fresh ECM medium was added to the lower chamber simultaneously. After incubating for 30 min at

37 °C in 5% $CO_2$, the fluorescence intensity in the lower chambers was measured using FlexStation 3 (Molecular Devices) at excitation/emission wavelength of 490/520 nm.

## CUT&Tag analysis

CUT&Tag analysis was performed using the Hyperactive Universal CUT&Tag Assay for Illumina Kit (Vazyme Biotech, TD904) according to the published protocol (Kaya-Okur et al, 2020). In brief, nuclei were isolated from $1 \times 10^5$ HAECs for each sample. Nuclei were light cross-linked by 0.1% formaldehyde for 2 min at room temperature and stopped by 75 mM glycine. Nuclei were collected using ConA Beads Pro and then incubated overnight at 4 °C with 0.5 μL of primary antibody (Mouse IgG, Abmart, B30010; Flag-tag, Abmart, M20008) diluted in 50 μL of antibody buffer. After incubation, beads were collected and incubated with 0.5 μL secondary antibody (Goat Anti-Mouse H&L, Vazyme Biotech, ab208-01) diluted in 50 μL dig-wash buffer for 1 h at room temperature. After three washes, beads were collected and incubated with 2 μL of pA/G-Tnp Pro diluted in 98 μL of Dig-300 Buffer for 1 h at room temperature. Beads were then washed three times and collected. Then, 50 μL of 1× TTBL diluted in Dig-300 Buffer was added to the beads and incubated for 60 min at 37 °C. Thereafter, 1 pg DNA spike-in was mixed with 2 μL of 10% SDS and added to every sample. The samples were then incubated for 10 min at 55 °C. The supernatant was then collected and incubated with DNA Extract Beads Pro for 20 min at room temperature. Next, the beads were collected and washed twice with B&W buffer. The beads were then resuspended in 20 μL of dd$H_2O$. Then, the sample was divided into two tubes of equal volume. One was for performing PCR to amplify the libraries, and the other was for performing qRT-PCR. Before performing qRT-PCR, samples were incubated with 5 μL of Stop Buffer for 5 min at 95 °C. The supernatant was collected as a template for qRT-PCR. All libraries were sequenced by NovaseqXP, and 150-bp paired-end reads were generated. Quality control was performed using an Agilent 2100. The data were analyzed using the cut_tag_tool developed by Vazyme (http://cloud.vazyme.com:83/).

## Plasmid transfection

293T cells were allowed to grow to 70–80% confluence before transfection. HiEff Trans liposomal transfection reagent (YEASEN, 40802ES03) was used for transfection, with all procedures following the manufacturer's instructions. In brief, 293T cells were transfected with the plasmid in Opti-MEM (Gibco, 31985088). The Opti-MEM was changed into complete medium at 6 h after transfection, and the cells were kept in culture for 48 h.

## DNA constructs and luciferase reporter assay

The CEBPB overexpression vector (cloned into pEnter vector, with C-terminal Flag and His tag) was obtained from WZ Biosciences Inc. (Shandong, China). For the luciferase reporter vector, the wild-type TGFBR1 promoter (−1409 to −570) and its six binding site mutants were synthesized and cloned into the pGL4.10 vector by SynbioB Co., Ltd. The quadruple mutant plasmid construct was generated through site-directed mutagenesis followed by cloning

into the pGL4.10 vector. For luciferase report assay, 293T cells were co-transfected with 1 μg of luciferase reporter vector, 1 μg of empty vector or CEBPB overexpression vector and 0.2 μg pRL-TK (Promega, E2241). Cells were cultured for 48 h and detected using the Dual Luciferase Reporter Gene Assay Kit (YEASEN, 11402ES60) following the manufacturer's protocol. The luciferase activity was assayed using a microplate reader (VICTOR Nivo, PerkinElmer).

## siRNA transfection

HAECs were allowed to grow to 70–80% confluence before transfection. HiEff Trans siRNA/miRNA reagent (YEASEN, 40806ES03) was used for transfection, with all procedures following the manufacturer's instructions. In brief, HAECs were transfected with siRNA at 50 nM in Opti-MEM (Gibco, 31985088). This medium was changed to a complete medium 6 h after transfection, and the cells were kept in culture for 24–48 h, followed by further treatment. The siRNA sequences used are as follows:

CEBPB siRNA, sense: AGCACAGCGACGAGUACAAGA; anti-sense: UUGUACUCGUCGCUGUGCUG

TGFBR1 siRNA, sense: CCAUCGAGUGCCAAAUGAATT; anti-sense: UUCAUUUGGCACUCGAUGGTT

## RNA sequencing and analysis

RNA sequencing was performed on HAECs in two groups, each of which contained three biological replicates. Cells were infected with adenovirus overexpressing CEBPB and a mock plasmid, as described above. The total RNA of each sample was then extracted using 350 μL TRIzol reagent (Ambion, 15596026), followed by RNA sequencing (LC-Bio Technologies (Hangzhou) Co., Ltd). StringTie and ballgown were used to estimate the expression levels of all transcripts and determine the expression abundance for mRNAs by calculating the fragment per kilobase of transcript per million mapped reads (FPKM) value. DEG analysis was performed using DESeq2 software between two different groups. The genes with the parameter of false discovery rate (FDR) below 0.05 and absolute fold change ≥2 were considered DEGs. The heatmap was generated using OmicStudio (https://www.omicstudio.cn/cell). GO analysis of differentially expressed genes and GSEA analysis were performed using the OmicStucio tools (https://www.omicstudio.cn/tool).

## Statistical analysis

Statistical analyses were performed using GraphPad Prism 9 (GraphPad software). Data were collected from at least three independent experiments and are presented as mean ± SD. For data with a sample size less than six, the normal distribution of data were checked by the Shapiro–Wilk test before statistical analysis. Statistical comparisons were performed using Student's *t*-test (two-group studies) or one-way analysis of variance (ANOVA) followed by Tukey's multiple comparison test (studies of more than two groups), as indicated. Exact *P* values are provided for all analyses, with values below 0.0001 displayed as <0.0001 according to GraphPad Prism 9's output style. Differences were considered statistically significant at *P* < 0.05.

## Data availability

All the data were available in the article and in its online Expanded View material. The data of single-cell RNA-seq, bulk RNA-seq, and CUT&Tag underlying this article are available in NCBI Gene Expression Omnibus (GEO), with the dataset identifiers GSE278346 (https://www.ncbi.nlm.nih.gov/geo/query/acc.cgi?acc=GSE278346), GSE278347 (https://www.ncbi.nlm.nih.gov/geo/query/acc.cgi?acc=GSE278347), and GSE278348 (https://www.ncbi.nlm.nih.gov/geo/query/acc.cgi?acc=GSE278348), respectively.

The source data of this paper are collected in the following database record: biostudies:S-SCDT-10_1038-S44318-025-00684-x.

## Peer review information

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

## Acknowledgements

We thank Dr. She-long Zhang and Dr. Fang-liang Huang (Equipment and Technology Service Platform, College of Life Sciences, Zhejiang University) for their excellent technical support with microscopy and flow cytometry. We thank the Equipment and Technology Service Platform of Zhejiang University School of Medicine for their excellent technical support with FACS sorting, and LC-Bio Technologies (Hangzhou) Co., Ltd for their technical assistance.

This work was supported by the National Key Research and Development Program of China [2021YFA1101100]; the National Natural Science Foundation of China [81970372, 82370450, 11932017, 91839104, 81770444, 81600354, and 31800972], Zhejiang Provincial Key R&D Program of China [2022C03097] and the Fundamental Research Funds for the Central Universities of China [K20220228].

## Author contributions

**Linge Fan**: Investigation; Writing—original draft; Project administration; Writing—review and editing. **Yingyi Zhu**: Investigation; Writing—original draft; Writing—review and editing. **Yi Li**: Investigation. **Zixin Ji**: Investigation. **Kefan Ma**: Investigation. **Ying Zhang**: Investigation. **Leiting Wei**: Investigation. **Junbo Chen**: Investigation. **Yuanqing Jiang**: Investigation. **Dongwu Lai**: Writing—review and editing. **Lingfeng Qin**: Writing—review and editing. **Guosheng Fu**: Writing—review and editing. **Michael Simons**: Resources; Writing—review and editing. **Liang Xu**: Investigation; Writing—review and editing. **Luyang Yu**: Supervision; Funding acquisition; Writing—review and editing. **Cong Qiu**: Data curation; Supervision; Funding acquisition; Project administration; Writing—review and editing.

Source data underlying figure panels in this paper may have individual authorship assigned. Where available, figure panel/source data authorship is listed in the following database record: biostudies:S-SCDT-10_1038-S44318-025-00684-x.

## Disclosure and competing interests statement

The authors declare no competing interests.

# Expanded View Figures

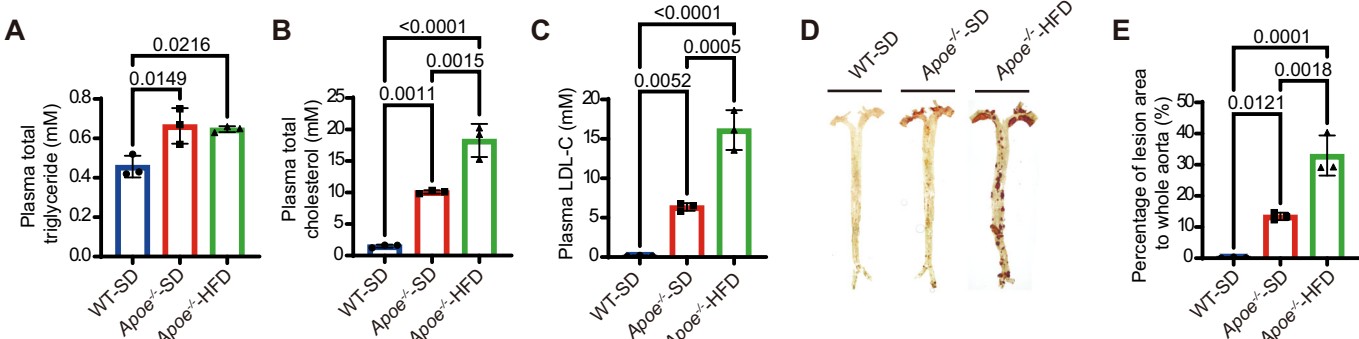

**Figure EV1. Lipid profile and lesion area of the WT-SD, $Apoe^{-/-}$-SD, and $Apoe^{-/-}$-HFD groups.**

(A) Plasma total triglyceride levels in the three groups ($n = 3$). (B) Plasma total cholesterol levels in the three groups ($n = 3$). (C) Plasma low-density lipoprotein cholesterol (LDL-C) levels in the three groups ($n = 3$). (D, E) Analysis of lesion area in the three groups. Representative images of Oil Red O staining of the whole aorta from the three groups are shown in (D), with quantification of the lesion area shown in (E) ($n = 3$). All data were presented as mean ± SD from three independent biological samples. Normal distribution was confirmed by the Shapiro–Wilk test. Statistical significance was determined by one-way ANOVA followed by Tukey's multiple comparison test. SD standard diet, HFD high-fat diet. $P < 0.05$ was considered statistically significant.

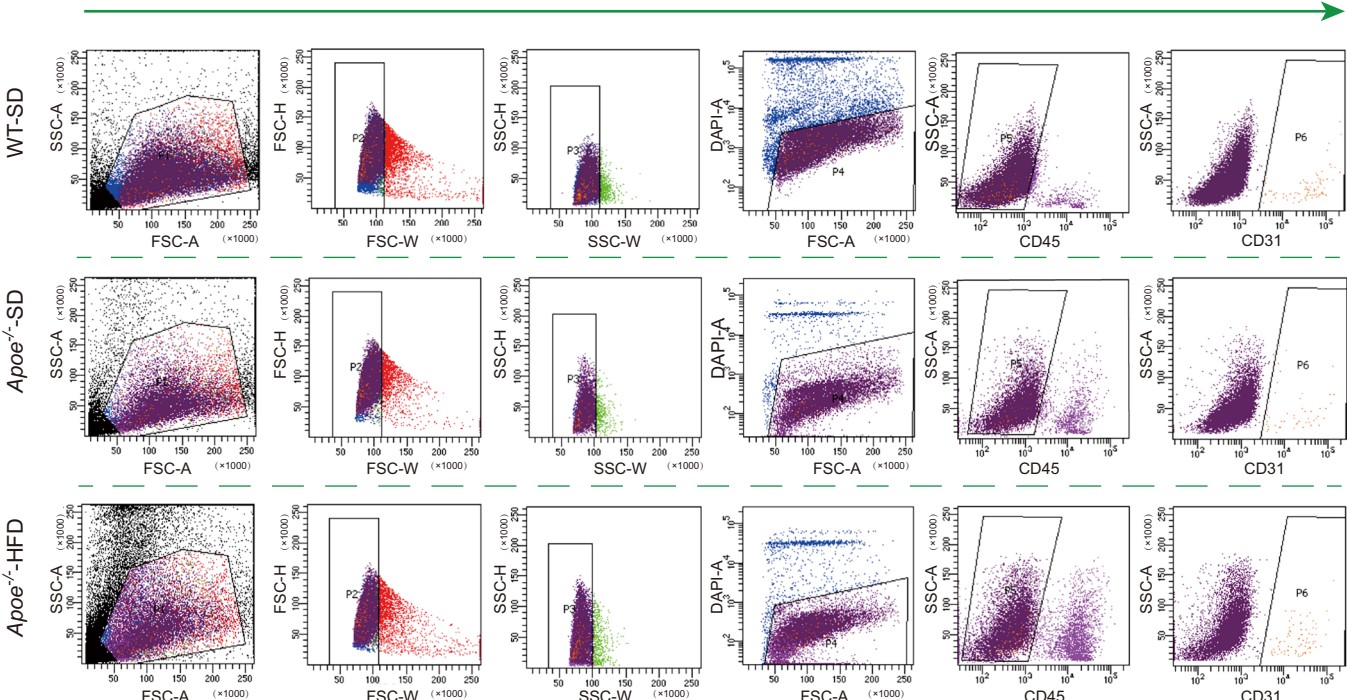

**Figure EV2. Gating strategy for FACS sorting.**

Doublets (i.e., P2 and P3) and dead cells (P4) were excluded from the total cells (P1). Endothelial cells were then identified as CD45$^-$ (P5) and CD31$^+$ (P6) cells. All cells for P6 were collected.

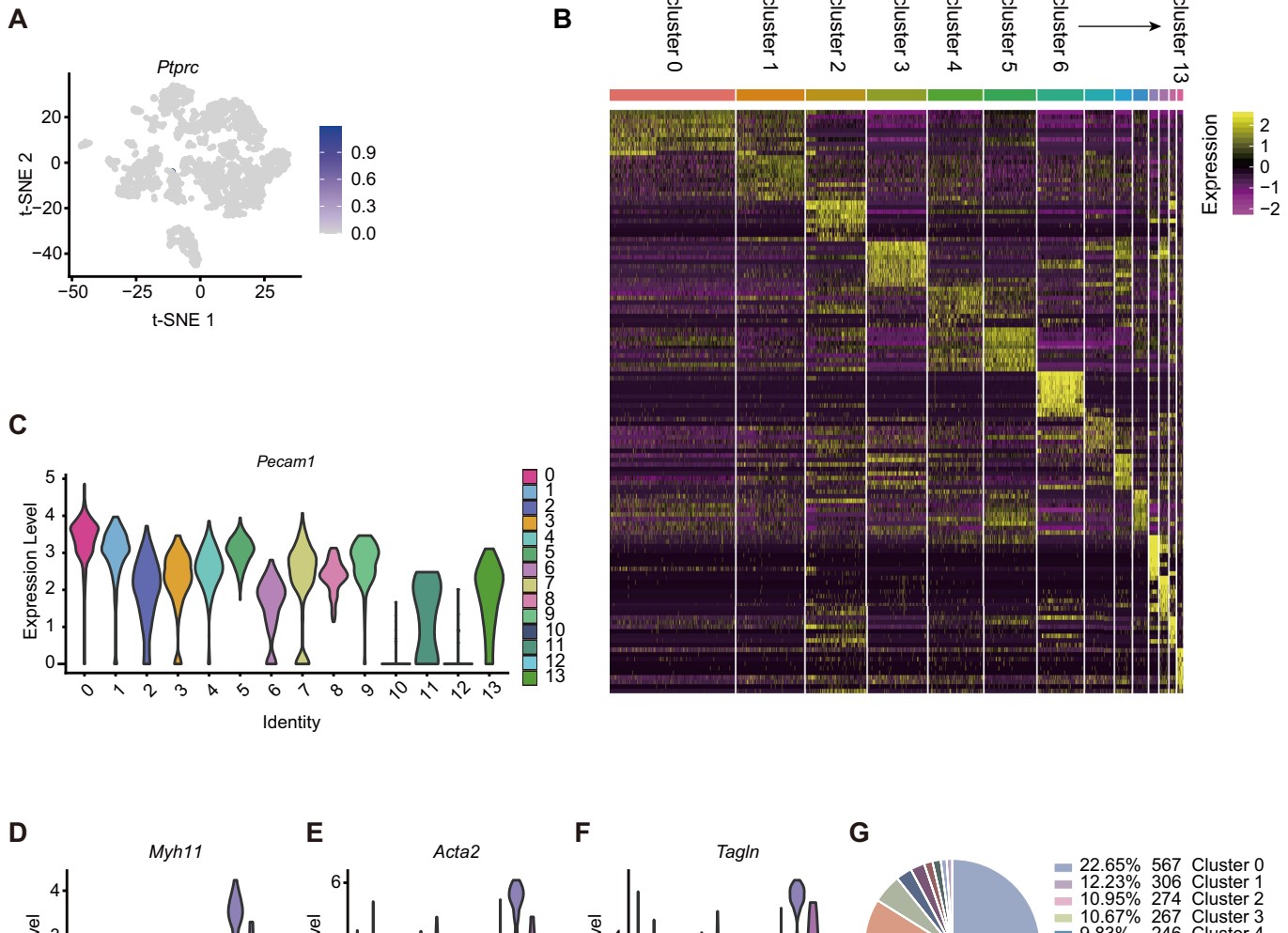

**Figure EV3. Gene expression levels for all clusters.**

(A) t-SNE plot of *Ptprc* (CD45) expression across all cells in the three groups. (B) Heatmap showing the top 10 marker genes from each cluster. (C) Violin plot of *Pecam1* expression for each cluster. (D) Violin plot of *Myh11* expression for each cluster. (E) Violin plot of *Acta2* expression for each cluster. (F) Violin plot of *Tagln* expression for each cluster. (G) Cell proportion (left column) and cell counts (middle column) for each cluster.

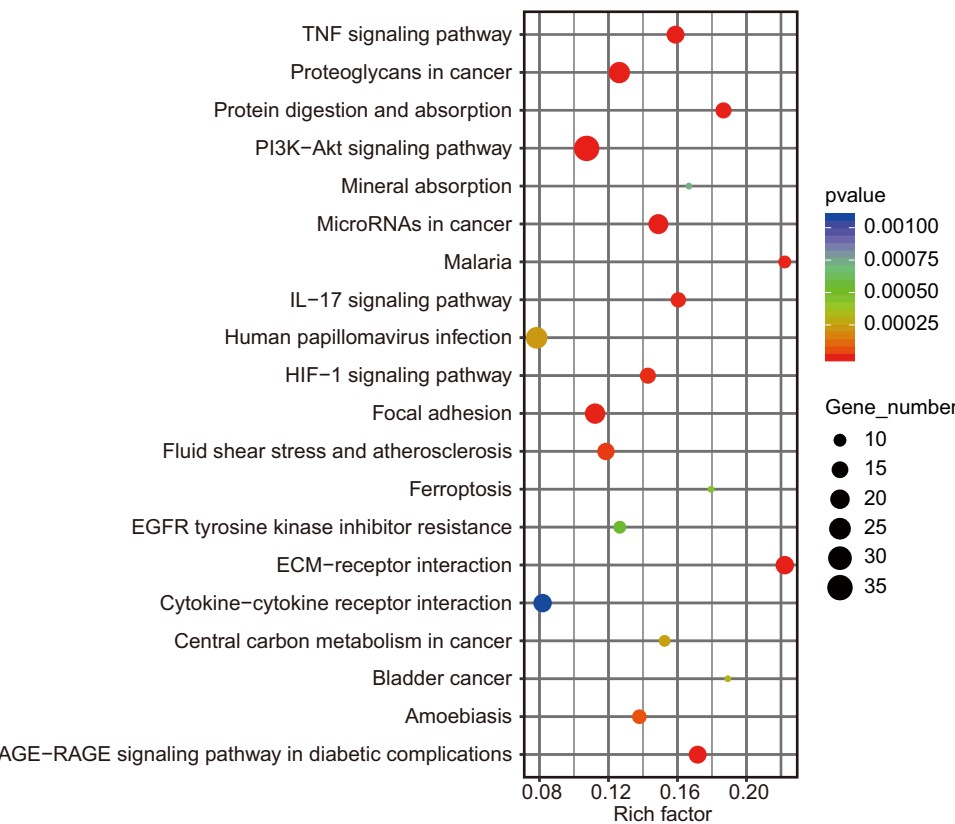

**Figure EV4. Top 20 KEGG pathways of cluster 2.**

KEGG enrichment analysis was performed using DEGs for cluster 2 relative to other clusters.

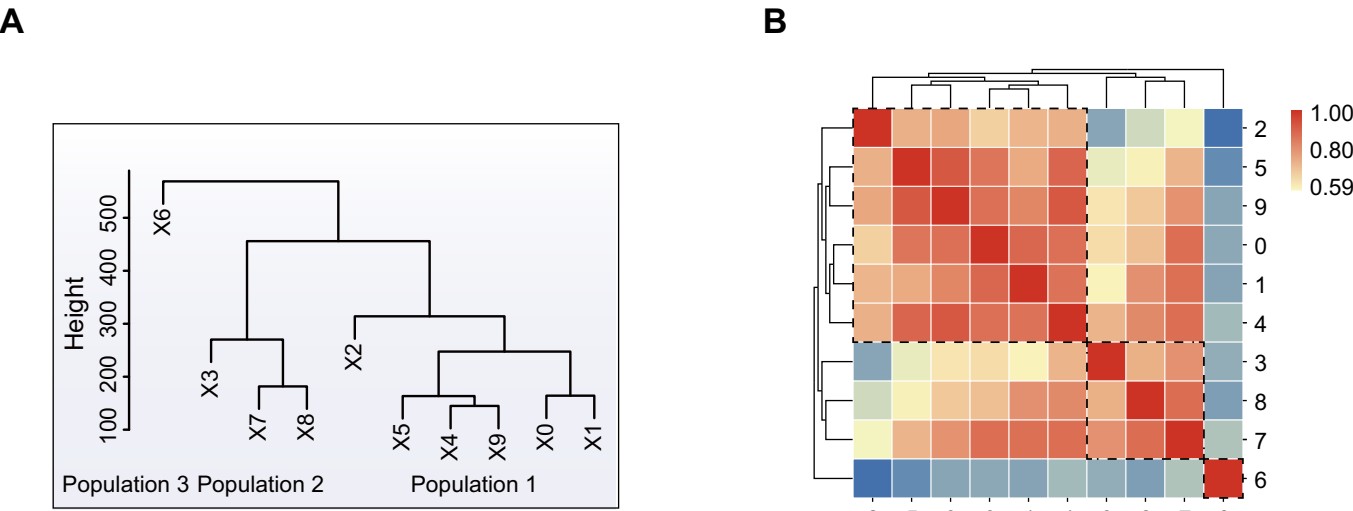

**Figure EV5. Correlation analysis among 10 clusters.**

(A) Dendrogram demonstrating the similarities of clusters 0–9 according to average RNA expression. "X" represents a cluster. (B) Pearson correlation analysis of ten EC clusters.

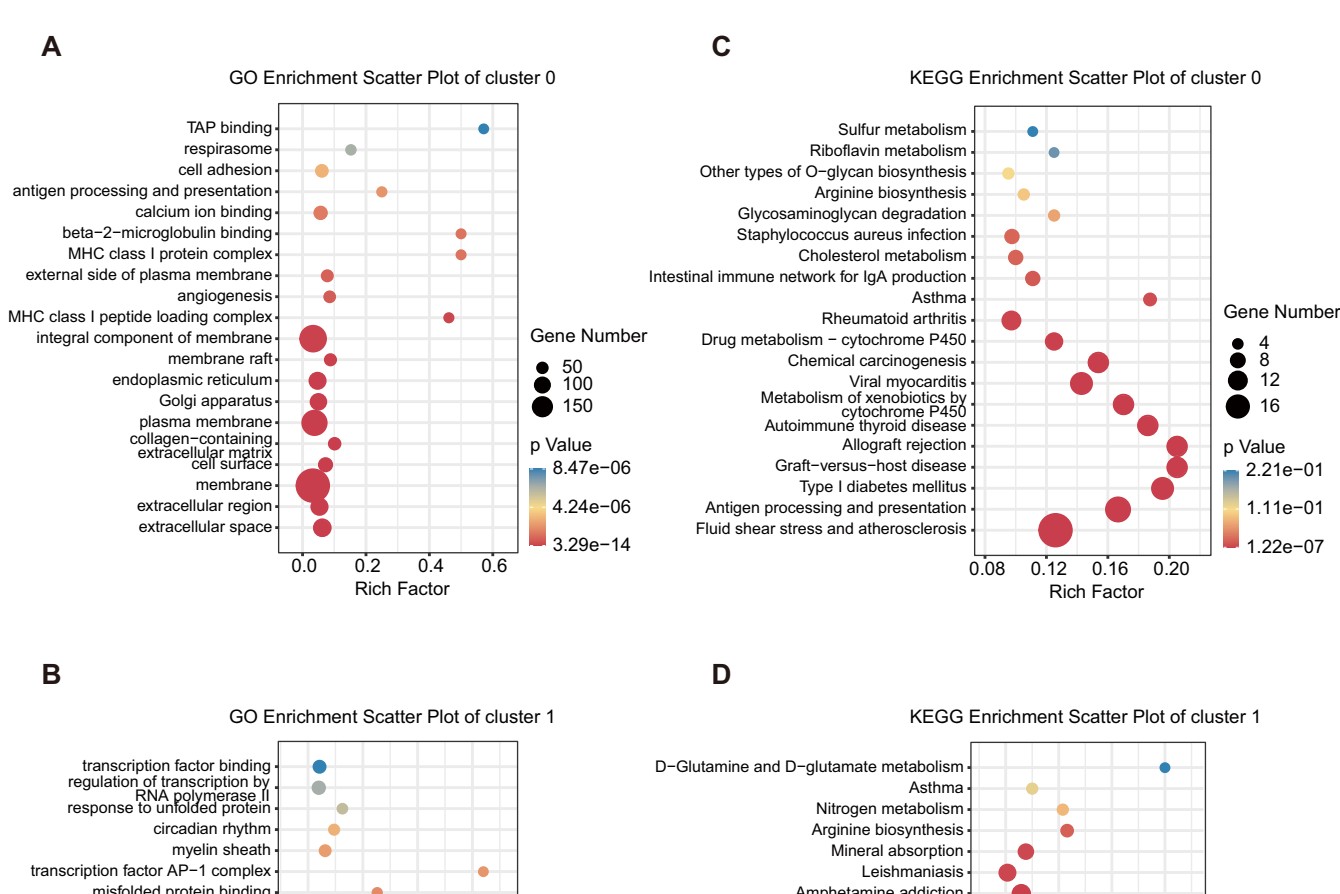

**Figure EV6. GO and KEGG enrichment of clusters 0 and 1.**

(A, B) Top 20 GO terms for cluster 0 (A) and cluster 1 (B). GO enrichment analysis was performed using DEGs for each cluster relative to other clusters. (C, D) Top 20 KEGG pathways for cluster 0 (C) and cluster 1 (D). The KEGG enrichment analysis was performed using DEGs for each cluster relative to other clusters.

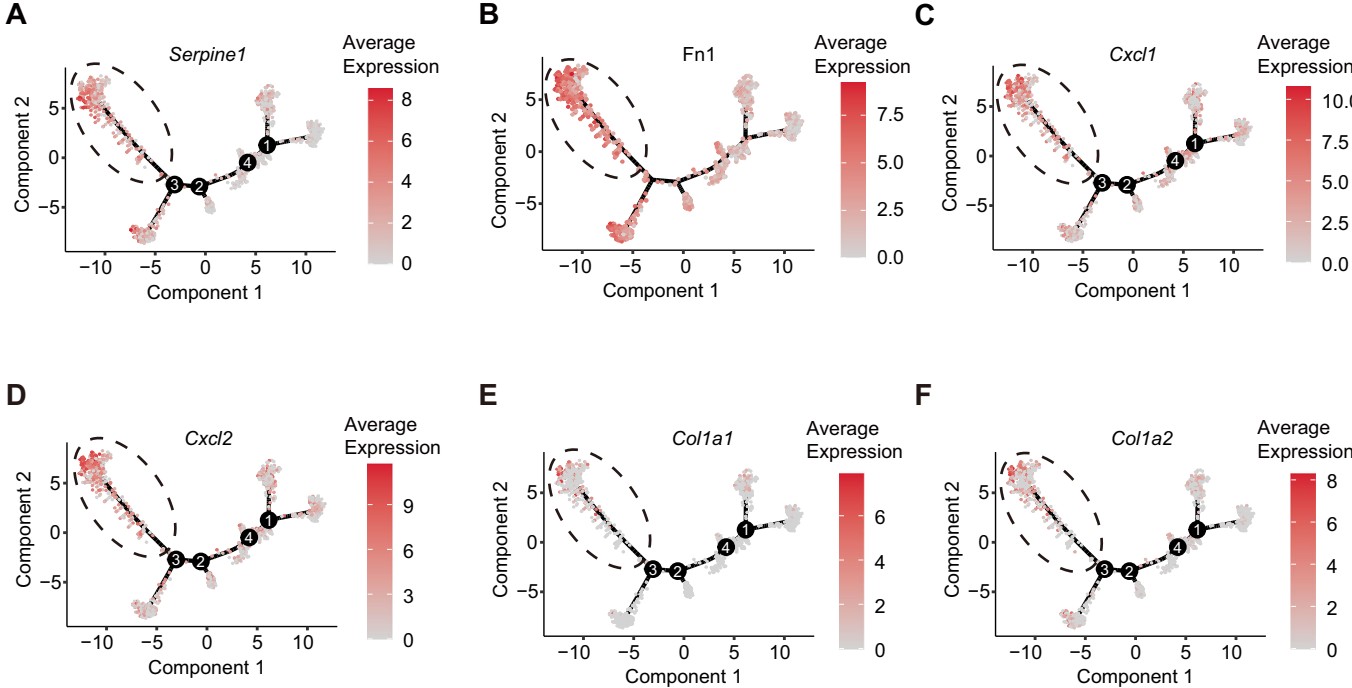

**Figure EV7. Gene trajectory analysis of fibroblast, inflammation, and ECM-related genes.**

The expression patterns of *Serpine1*, *Fn1*, *Cxcl1*, *Cxcl2*, *Col1a1*, and *Col1a2* along the pseudotime trajectory are shown in (**A–F**), respectively.

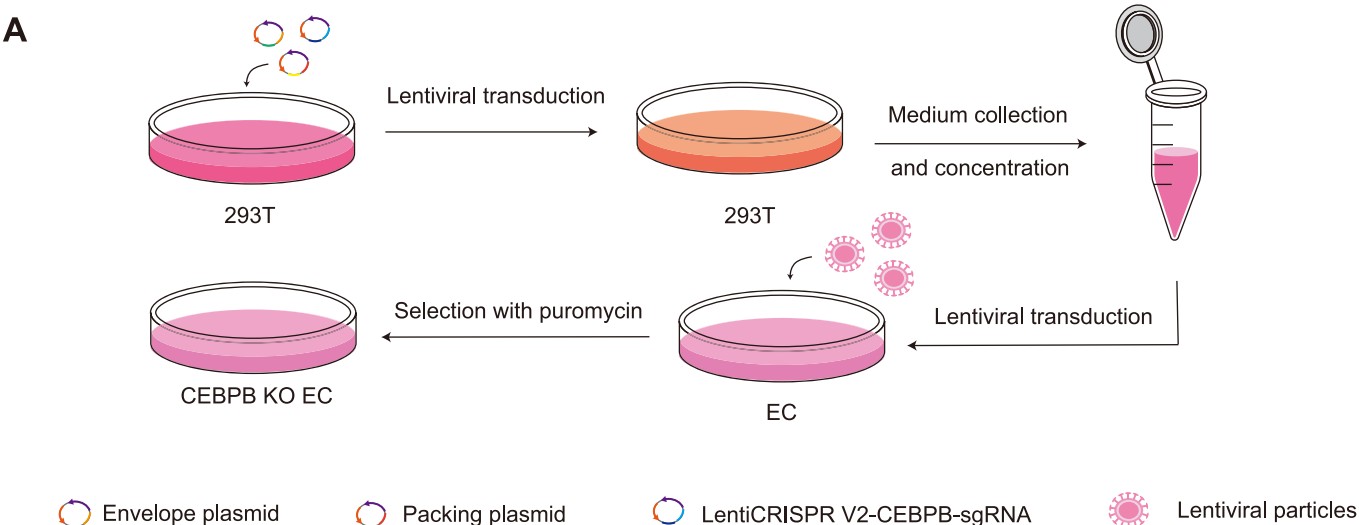

**A**

**B**

| CEBPB sg1-F | CACCGCGCTTACCTCGGCTACCAGG |
|---|---|
| CEBPB sg1-R | AAACCCTGGTAGCCGAGGTAAGCGC |
| CEBPB sg2-F | CACCGAGTACGGCTACGTGAGCCTG |
| CEBPB sg2-R | AAACCAGGCTCACGTAGCCGTACTC |
| CEBPB sg3-F | CACCGTCGACTTCAGCCCGTACCTG |
| CEBPB sg3-R | AAACCAGGTACGGGCTGAAGTCGAC |

**C**

Figure EV8.  Validation of C/EBPβ knockout in HAECs.

(A) Schematic diagram of C/EBPβ knockout using CRISPR-Cas9. (B) Sequence of three independent sgRNA pairs targeting C/EBPβ. (C) Western blot analysis of C/EBPβ protein expression following CRISPR/Cas9-mediated knockout with three independent sgRNAs as indicated. All blots were performed on separate membranes and were not subjected to reprobing.

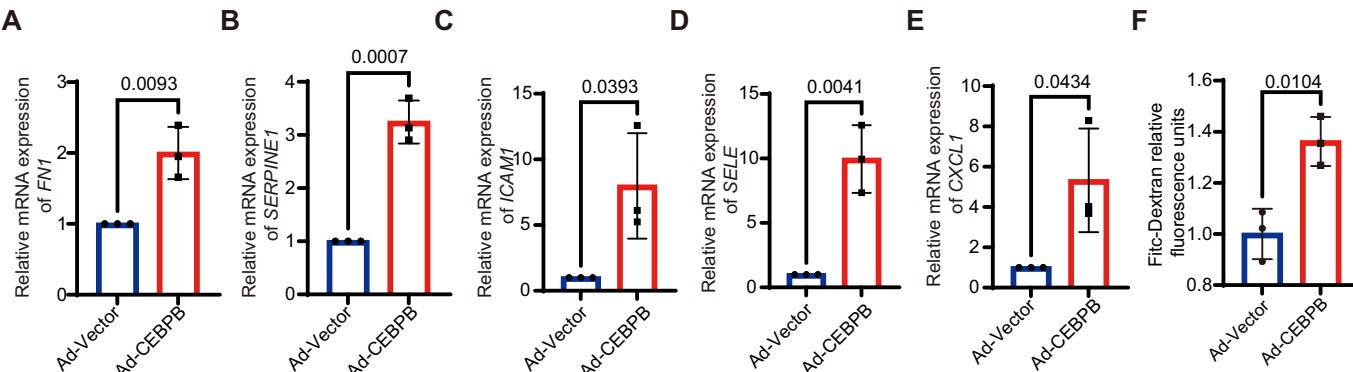

**Figure EV9.  C/EBPβ overexpression promotes fibroblast-like EC phenotype.**

(A–E) Relative mRNA level of *FN1*, *SERPINE1*, *ICAM1*, *SELE*, and *CXCL1* in HAECs infected with Ad-Vector or Ad-C/EBPβ (*n* = 3). (F) Relative value of FITC-Dextran fluorescence units from HAECs infected with Ad-Vector or Ad-C/EBPβ (*n* = 3). Values in are shown as mean ± SD. Normal distribution was confirmed by the Shapiro–Wilk test. Differences were analyzed by an unpaired *t*-test (two-tailed). *P* < 0.05 was considered statistically significant.

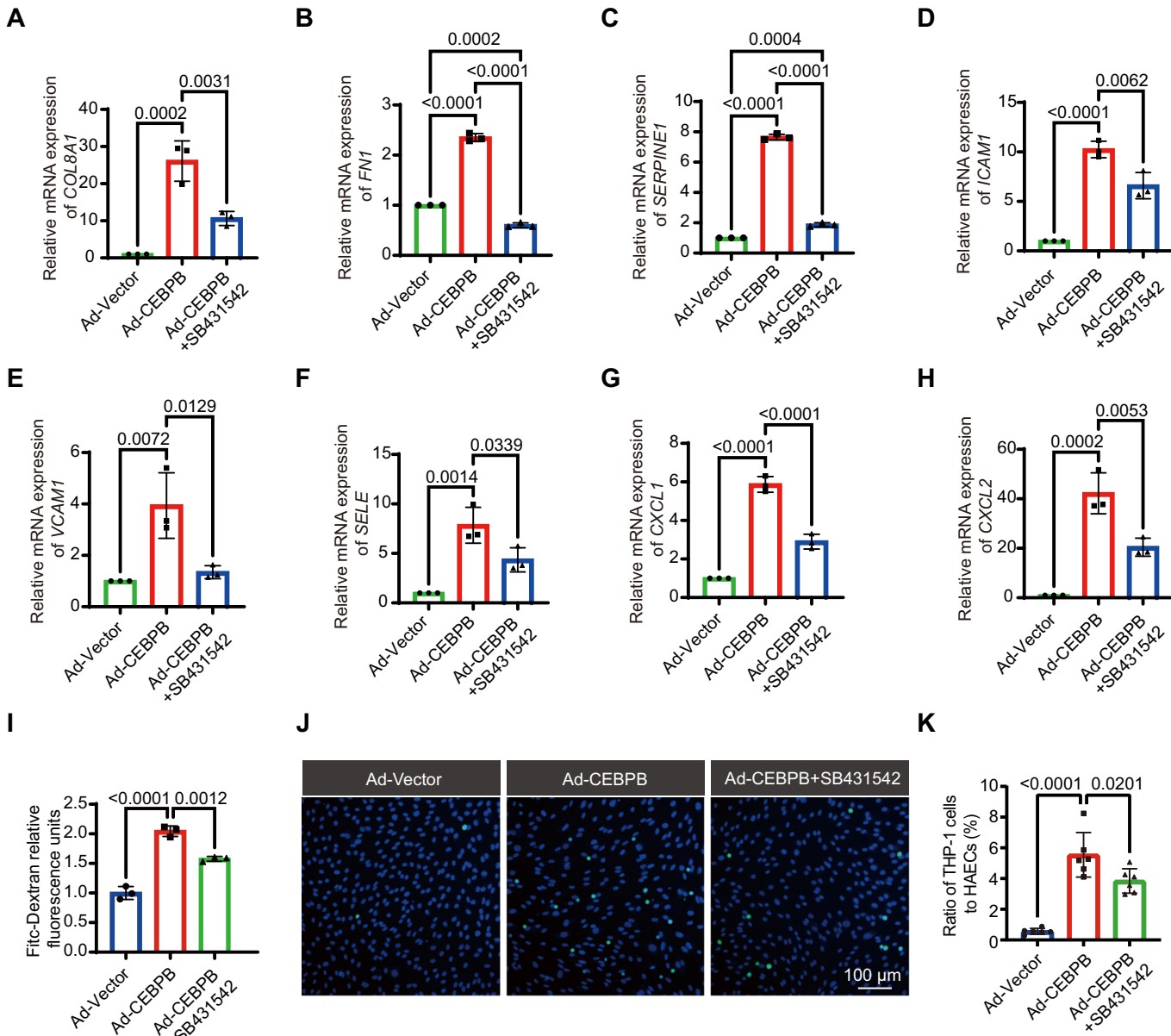

**Figure EV10.    TGF-β signaling inhibitor SB431542 suppressed endothelial phenotypic switching and inflammation induced by Ad-C/EBPβ.**

(A–H) Relative mRNA expression of *COL8A1, FN1, SERPINE1, ICAM1, VCAM1, SELE, CXCL1,* and *CXCL2* in HAECs (*n* = 3). Cells were infected with Ad-Vector or Ad-CEBPB and cultured in full medium with 10 μM of SB431542 or the vehicle for 72 h before harvest. (I) Relative value of FITC-Dextran fluorescence units of HAECs. Cells were infected with Ad-Vector or Ad-C/EBPβ and cultured in full medium with 10 μM of SB431542 or the vehicle for 72 h before FITC-dextran treatment (*n* = 3). (J, K) SB431542 reduced the adhesion of THP-1 to HAECs caused by Ad-C/EBPβ. Representative images with adherent THP-1 cells (green) are shown in (J), with the relative quantification of the ratio of THP-1 cells to HAECs shown in (K) (*n* = 6). Scale bar, 100 μm. Values are presented as mean ± SD. Normal distribution was confirmed by the Shapiro–Wilk test. Statistical significance was determined by one-way ANOVA followed by Tukey's multiple comparison test. *P* < 0.05 was considered statistically significant.

**A**

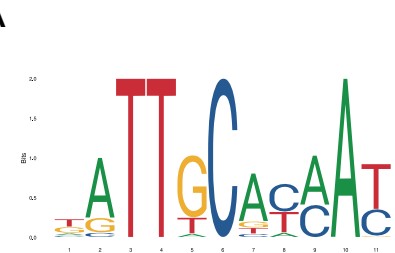

| Relative score | Binding site |
|---|---|
| 0.8573209 | -1382~-1366 |
| 0.82612014 | -1075~-1065 |
| 0.82415706 | -983~-973 |
| 0.8214078 | -609~-599 |
| 0.80399895 | -1269~-1259 |
| 0.80344266 | -1149~-1139 |

**C**

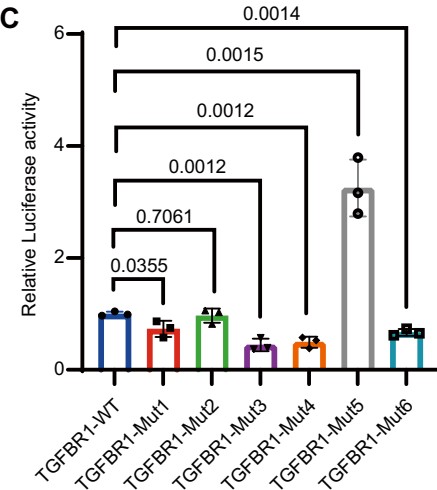

**B**

(-1382~-1366)   luciferase   TGFBR1-WT:   ACTTGCTTAAGTAAGT
TGFBR1-Mut1: CAGGTAGGCCTGCCTG

(-1269~-1259)   luciferase   TGFBR1-WT:   GTGGGGAAAAT
TGFBR1-Mut2: TGTTTTCCCCG

(-1149~-1139)   luciferase   TGFBR1-WT:   AATTTCCTCCT
TGFBR1-Mut3: CCGGGAAGAAG

(-1075~-1065)   luciferase   TGFBR1-WT:   CTGAAGAAAGG
TGFBR1-Mut4: AGTCCTCCCTT

(-983~-973)   luciferase   TGFBR1-WT:   ATTAGGAATTC
TGFBR1-Mut5: CGGCTTCCGGA

(-609~-599)   luciferase   TGFBR1-WT:   AGGAGGAAATA
TGFBR1-Mut6: CTTCTTCCCGC

**D**

TGFBR1-4Mut
(Mut1,Mut3,Mut4,Mut6)

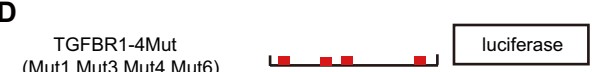   luciferase

**Figure EV11.    Identification of C/EBPβ binding motif.**

(A) Predicted sites of the putative C/EBPβ binding motif. C/EBPβ binding elements in the promoter region of *TGFBR1* (from −1409 to −570) were predicted using the JASPAR database, and the six most likely binding sites (relative score >0.8) were selected based on the predicted scores. (B) Schematic diagram of mutant construction. The six mutants were respectively fused in frame to the pGL4.10 luciferase reporter vector and named TGFBR1-Mut1, TGFBR1-Mut2, TGFBR1-Mut3, TGFBR1-Mut4, TGFBR1-Mut5, and TGFBR1-Mut6. The bases A, T, C, and G at the sites were mutated to C, G, A, and T, respectively. (C) Dual luciferase assay indicating the binding sites of C/EBPβ for *TGFBR1* transcription in 293T cells (n = 3). Relative luciferase activity was normalized to Renilla and the TGFBR1-WT group. Values are shown as mean ± SD. Normal distribution was confirmed by the Shapiro–Wilk test. All differences were analyzed using an unpaired *t*-test (two-tailed). $P < 0.05$ was considered statistically significant. (D) Schematic diagram of the construction of the quadruple mutant plasmid. Based on the results presented in (C), four potential binding sites were mutated to construct a quadruple mutant plasmid, which was named TGFBR1-4Mut.

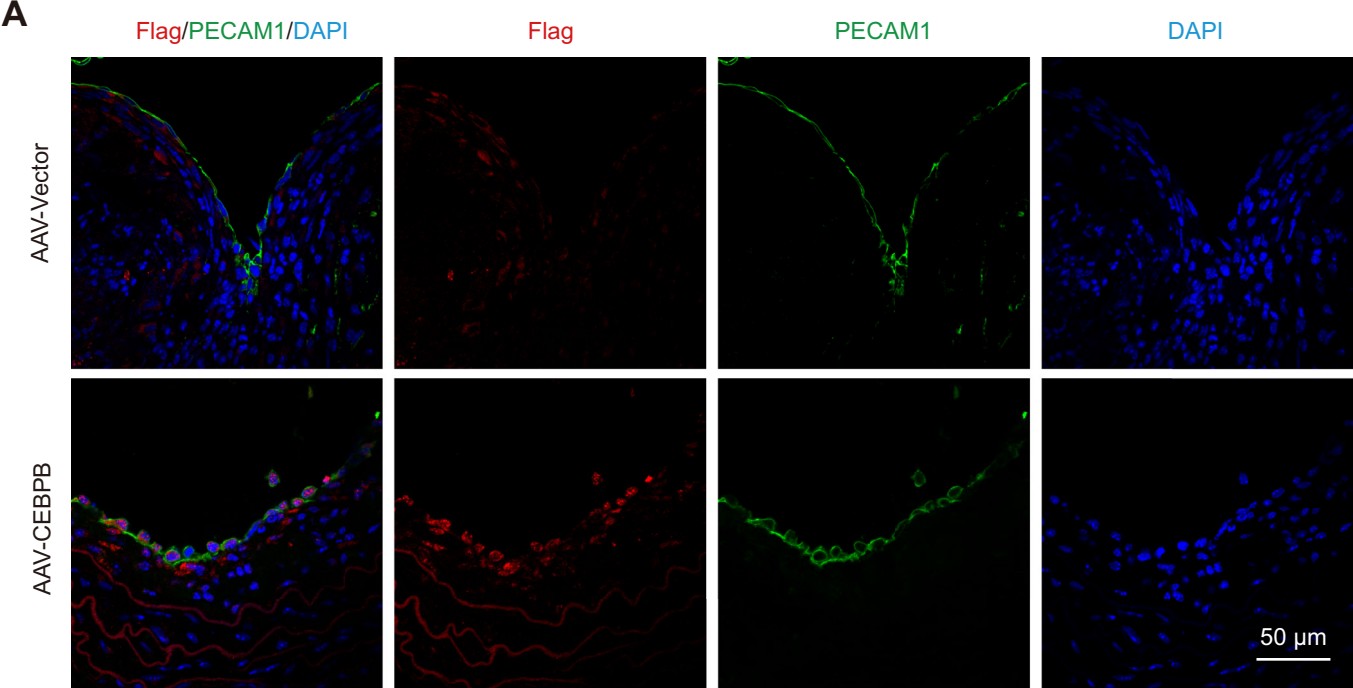

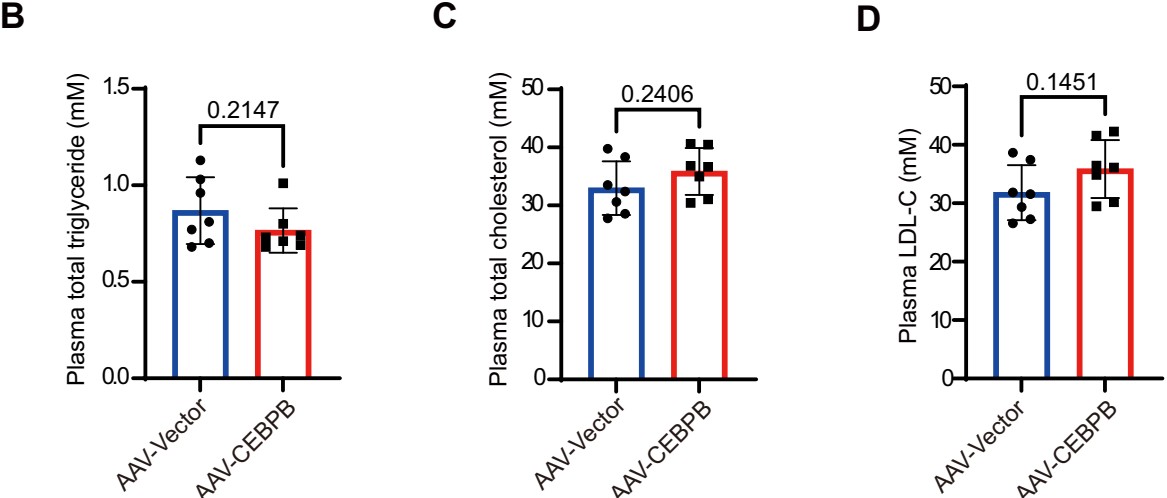

**Figure EV12.  AAV-C/EBPβ does not affect lipid profile.**

(A) Representative IF staining of Flag (C/EBPβ, red), PECAM1 (green), and DAPI (blue) in arteries from AAV-Vector and AAV-CEBPB mice. Scale bar, 50 μm. (B) Plasma total triglyceride levels in AAV-Vector and AAV-C/EBPβ mice ($n = 7$). (C) Plasma total cholesterol levels in AAV-Vector and AAV-C/EBPβ mice ($n = 7$). (D) Plasma LDL-C levels in AAV-Vector and AAV-C/EBPβ mice ($n = 7$). Plasma levels of total triglyceride, total cholesterol, and LDL-C were measured after 8 weeks of HFD feeding. Values are presented as mean ± SD. Statistical significance was determined by an unpaired $t$-test (two-tailed). $P < 0.05$ was considered statistically significant.

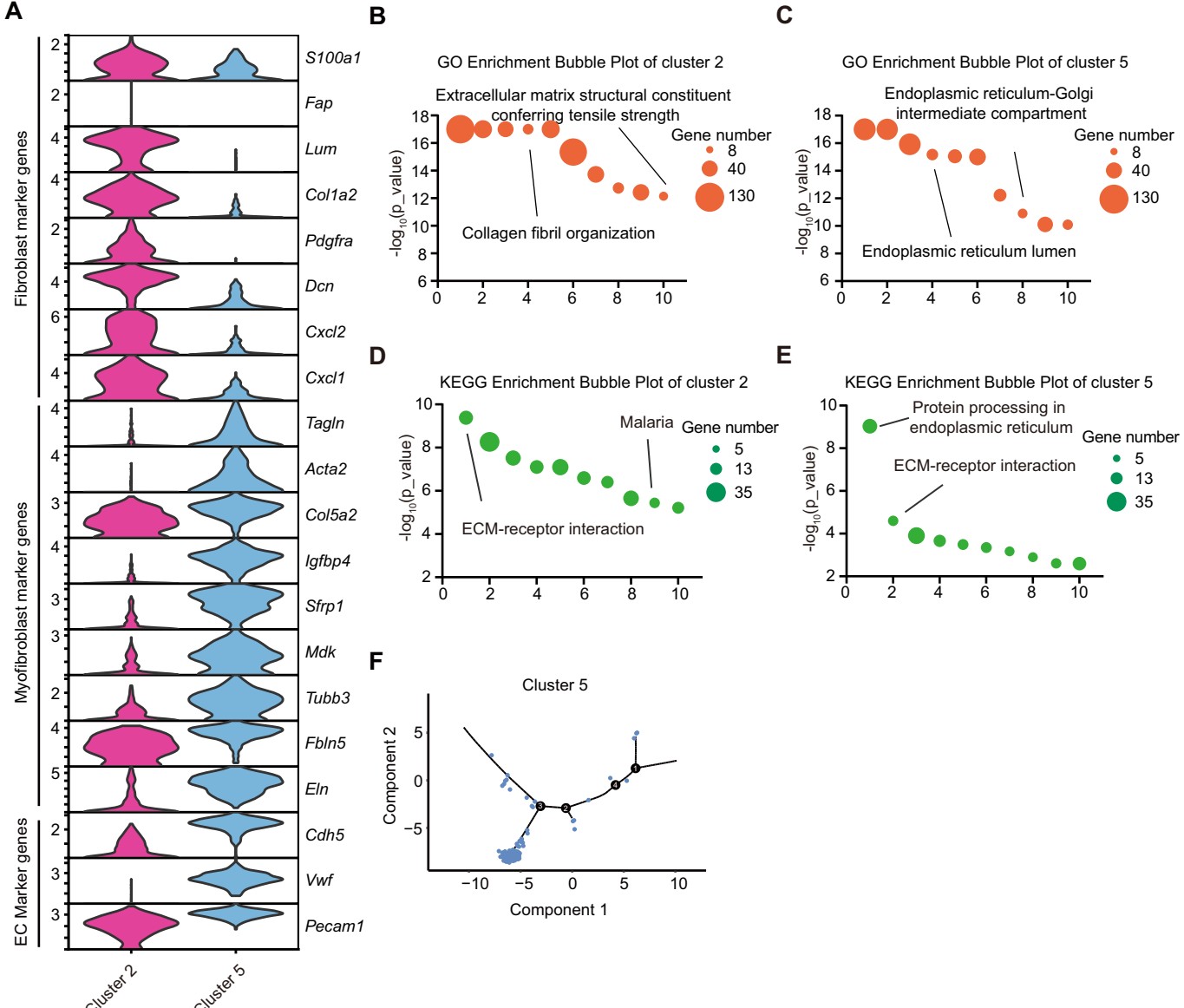

**Figure EV13.  Clusters 2 and 5 represent different cell fates.**

(A) Violin plot of selected marker genes from clusters 2 and 5. (B, C) Top 10 GO terms for cluster 2 (B) and cluster 5 (C). GO enrichment analysis was performed on DEGs from each cluster relative to all other clusters. GO terms with the top two enrichment factors are marked on the graphs. (D, E) Top ten KEGG pathways present in cluster 2 (D) and cluster 5 (E) cells. KEGG enrichment analysis was performed on DEGs for each cluster relative to other clusters. KEGG pathways with the top two highest levels of enriched components are marked on the graphs. (F) Monocle pseudotime trajectory analysis of cluster 5.

