## [Peer Review File · The EMBO Journal]

A fibroblast-like endothelial cell state promotes atherosclerosis via C/EBP β -activated TGF- β signaling

Linge Fan, Yingyi Zhu, Yi Li, Zixin Ji, Kefan Ma, Ying Zhang, Leiting Wei, Junbo Chen, Yuanqing Jiang, Dongwu Lai, Lingfeng Qin, Guosheng Fu, Michael Simons, Liang Xu, Luyang Yu, and Cong Qiu

Corresponding authors: Cong Qiu (congqiu@zju.edu.cn) , Luyang Yu (luyangyu@zju.edu.cn)

Review Timeline:

Submission Date:	25th Apr 25
Editorial Decision:	7th Jun 25
Revision Received:	5th Oct 25
Editorial Decision:	4th Dec 25
Revision Received:	10th Dec 25
Accepted:	16th Dec 25

Editor: Daniel Klimmeck

Transaction Report:

Dear Dr Qiu,

Thank you for submitting your manuscript EMBOJ-2025-121133 to The EMBO Journal, as well as for your patience with our feedback at this time of the year. Your manuscript was sent to three reviewers with expertise in endothelial cell biology, vascular disease and single-cell analyses, and we have now received reports from two of them, which I enclose below. Please note that referee #1 with expertise on single-cell approaches has not sent us his/her report yet; however, we have in the interest of time now decided to base our decision on the other existing comments. Accordingly, we can invite you to revise your work for the EMBO Journal.

As you will see from the experts' reports, they acknowledge the potential interest and value of your findings. However, they also point to important shortcomings with the analysis, which need to be rigorously addressed to make them supportive of publication in the EMBO Journal. In particular, key issues are raised regarding lack of loss-of-function approaches to support endogenous relevance of the CEBPB-TGFBR1 axis; proof of specificity of the AAV-based overexpression experiments; completeness of the bioinformatic cluster analyses. Further, the reviewers raise a number of complementary issues related to the presentation of the findings, and data annotation, as well as overall writing of the manuscript that would need to be conclusively addressed to achieve the level of robustness and clarity needed for The EMBO Journal.

Given the overall interest stated and broader angle of your findings, we are able to invite you to revise your manuscript experimentally to address the referees' comments, pending there are no overriding technical argument brought forward by the referee #1. I will let you know as soon as we have received this expert's feedback.

Overall, I need to stress that we do require strong support from the referees on a revised version of the study in order to move on to publication of the work.

Please feel free to contact me if you have any questions or need further input on the referee comments.

When submitting your revised manuscript, please carefully review the instructions below.

Please feel free to approach me any time should you have additional questions related to this.

Thank you for the opportunity to consider your work for publication.

I look forward to your revision.

Kind regards,

Daniel Klimmeck

Daniel Klimmeck, PhD
Senior Editor
The EMBO Journal

Instruction for the preparation of your revised manuscript:

1) a .docx formatted version of the manuscript text (including legends for main figures, EV figures and tables). Please make sure

that the changes are highlighted to be clearly visible.

2) individual production quality figure files as .eps, .tif, .jpg (one file per figure).

3) a .docx formatted letter INCLUDING the reviewers' reports and your detailed point-by-point response to their comments. As part of the EMBO Press transparent editorial process, the point-by-point response is part of the Review Process File (RPF), which will be published alongside your paper.

4) a complete author checklist, which you can download from our author guidelines ([https://wol-prod-cdn.literatumonline.com/pb-assets/embo-site/Author Checklist%20-%20EMBO%20J-1561436015657.xlsx](https://wol-prod-cdn.literatumonline.com/pb-assets/embo-site/Author%20Checklist%20-%20EMBO%20J-1561436015657.xlsx)). Please insert information in the checklist that is also reflected in the manuscript. The completed author checklist will also be part of the RPF.

6) It is mandatory to include a 'Data Availability' section after the Materials and Methods. Before submitting your revision, primary datasets produced in this study need to be deposited in an appropriate public database, and the accession numbers and database listed under 'Data Availability'. Please remember to provide a reviewer password if the datasets are not yet public (see <https://www.embopress.org/page/journal/14602075/authorguide#datadeposition>).

7) Our journal encourages inclusion of *data citations in the reference list* to directly cite datasets that were re-used and obtained from public databases. Data citations in the article text are distinct from normal bibliographical citations and should directly link to the database records from which the data can be accessed. In the main text, data citations are formatted as follows: "Data ref: Smith et al, 2001" or "Data ref: NCBI Sequence Read Archive PRJNA342805, 2017". In the Reference list, data citations must be labeled with "[DATASET]". A data reference must provide the database name, accession number/identifiers and a resolvable link to the landing page from which the data can be accessed at the end of the reference. Further instructions are available at .

8) At EMBO Press we ask authors to provide source data for the main and EV figures. Our source data coordinator will contact you to discuss which figure panels we would need source data for and will also provide you with helpful tips on how to upload and organize the files.

Numerical data can be provided as individual .xls or .csv files (including a tab describing the data). For 'blots' or microscopy, uncropped images should be submitted (using a zip archive or a single pdf per main figure if multiple images need to be supplied for one panel). Additional information on source data and instruction on how to label the files are available at .

9) We replaced Supplementary Information with Expanded View (EV) Figures and Tables that are collapsible/expandable online (see examples in <https://www.embopress.org/doi/10.15252/emboj.201695874>). A maximum of 5 EV Figures can be typeset. EV Figures should be cited as 'Figure EV1, Figure EV2' etc. in the text and their respective legends should be included in the main text after the legends of regular figures.

11) For data quantification: please specify the name of the statistical test used to generate error bars and P values, the number (n) of independent experiments (specify technical or biological replicates) underlying each data point and the test used to

calculate p-values in each figure legend. The figure legends should contain a basic description of n, P and the test applied. Graphs must include a description of the bars and the error bars (s.d., s.e.m.).

We realize that it is difficult to revise to a specific deadline. In the interest of protecting the conceptual advance provided by the work, we recommend a revision within 3 months (5th Sep 2025). Please discuss the revision progress ahead of this time with the editor if you require more time to complete the revisions.

Referee #2:

The study by Fan et al., offers a mechanistic exploration of endothelial cell (EC) plasticity in atherosclerosis. Using single-cell RNA sequencing, the authors identify a fibroblast-like EC population characterized by EndMT, inflammation, and ECM-remodeling signatures, which expands with disease severity. CEBPB is identified as the dominant transcription factor driving this phenotype, inducing expression of TGFBR1, promoting TGF- β signaling. Overexpression of CEBPB in ECs in vivo leads to worsened atherosclerotic lesions, supporting an important role for CEBPB in driving endothelial dysfunction. The authors propose the CEBPB-TGFBR1 axis as a novel mechanistic and therapeutic target in atherosclerosis. The authors exploit a range of state-of-the-art in vitro and in vivo methodology in an overall well-presented paper with a commendable description of limitations of the study.

Major comments

1. The authors use scRNAseq to explore gene expression changes in endothelial cells isolated from wildtype aorta, or aortas from Apoe^{-/-} mice on standard diet (SD) or on high fat diet (HFD). FACS-isolated ECs (CD45⁻, CD31⁺) are resolved in 13 clusters, of which the authors are particularly interested in cluster 2. This cluster stands out with increased expression of fibroblastoid markers (Lum, Dcn1) and essentially lacking typical endothelial markers (ERG, Cdh5). Cluster 2 is described as expanding with increased atherosclerotic lesions. However, in Supple Table 3, cluster 2 proportion is 10.64% in WT-SD, 16.4% in Apoe^{-/-} SD and 9.08% in Apoe^{-/-} HFD ECs. In contrast, several other clusters such as 4, 5 and 8 do not exist at all in the WT SD condition. The authors need to explain how cluster 2 can drive disease if the proportion is similar between the healthy and the atherosclerotic aorta.
2. Cluster 2 ECs express several fibroblast markers. Their seemingly perivascular localization (see Fig. 3 panels D and E) would also be compatible with an inflammatory cell type. Do cluster 2 cells express typical inflammatory cell markers such as CD68, CD163, Itgam, Adgre1 (F4/80)? Can the authors exclude that cluster 2 consists of co-purified inflammatory cells which also express CD31? Many of the in vivo findings could be explained by assuming that this is a co-purified population of fibroblasts and/or inflammatory cells. Please note the study by Ma et al., 10.1038/s41392-025-02196-w entitled "C/EBP β activation in vascular smooth muscle cells promotes hyperlipidemia-induced phenotypic transition and arterial stiffness". This paper should be cited.
3. To address the relationship between the different clusters (shown in Fig. 2), the authors perform pseudotime trajectory analyses where their three different mouse strains/conditions are compared. It is not explained how this analysis was done, please add to Methods. However, rather than comparing the three strains, it would be preferable to examine trajectories of one and the same strain over time, for example the Apoe^{-/-} strain given the HFD for different time periods.
4. In Fig. 3, panels D and E, the authors immunostain both human and mouse atherosclerotic plaques to show expression of CEBPB and CD31. The CEBPB signal increases with atherosclerosis but there is hardly any overlap with CD31? Please quantify the co-localized area in the two conditions (which need to include at least 3 samples/human and condition). Also, the labels are confusing, showing - and + for human or mouse "atherosclerotic plaque". What is the - (minus) sample? Please describe better.
5. In Fig. 4, the authors measure transcript levels in HAECs infected with control or Ad.CEBPB; please include also Lum and Dcn1.

Minor comments

6. On line 161, the authors mention a "rapid decrease in EC fate markers". What does "rapid" refer to; kinetics have not been determined?

7. On line 266 the authors describe CEBPB as a "novel transcription factor of TGFBR1". As CEBPB has been implicated in regulation of TGFb expression the authors' statement could be toned down and the paper by Ma et al (see above) should be referenced here.

8. Even though the authors overall manage the English language quite well, there are errors which attract unnecessary attention, therefore, professional language revision is recommended.

Referee #3:

In this manuscript, the authors compared aortic gene expression of WT mice and ApoE mice with and without high fat diet. scRNAseq yielded 14 endothelial clusters, of which one uniquely expressed fibroblast associated genes (Lum, Dcn). In these cells, the transcription factor CEBPB was selectively upregulated. Exposure of human aortic endothelial cells to IL-1beta and TNF-alpha increased CEBPB expression. Adenoviral expression of CEBPB increased marker genes of cluster 2 endothelial cells and induced a fibroblast like phenotype. GSEA point towards TGF-beta signaling and indeed, SMAD2 was phosphorylated in response to CEBPB and inhibition of ALK reduced the effects on phenotype. A potential explanation for this was an induction of TGFBR1 by CEBPB. Was demonstrated by CUT&Tag, CEBPB bound the TGFbeta receptors. Overexpression of CEBPB also promoted atherosclerosis development in mice.

This is a nice manuscript suggesting some role of CEBPB in atherosclerosis development. Unfortunately, as no loss of function experiments were performed, ultimately, the relevance of CEPBP remains unclear. This reviewer has the following comments:

Major comments:

- 1) Loss of function experiments are required to address the endogenous function of CEBPB. Given that there is often redundancy in the action of TFs, such experiments are needed to demonstrate that CEBPB is also relevant.
- 2) Which AAV-Serotype was used? How was endothelial specific expression achieved beyond the Tie2 promoter
- 3) Fig 3B: what are the other up and down-regulated TFs? Please provide a table in the supplements.
- 4) Fig 3B yields at least 10 interesting TFs (4 down, 6 up). What are these? Why did the authors exclude them from further analysis?
- 5) Fig 4A: Is this the complete list of differentially regulated genes? Dcn and lum are missing. Why is that so?
- 6) Fig 5Q: Which other promoters were also bound by CEBPB? Did they also coincide with gene expression alterations in endothelial cells? How many of which of the cluster 2 gene promoters were bound by CEBPB?
- 7) Fig 5Q. Along the same line: Was there any binding to enhancers?

Minor comments:

- 1) Please provide the reviewer password to review the GEO deposited data
- 2) Page 7, line 190&193&194: This statement is invalid here. Only a coincidence is reported not a causality.

Dear Editors and Referees,

We appreciate the opportunity to revise our manuscript titled "**The CEBPB-TGFBR1 axis drives fibroblast-like endothelial cell phenotype and promotes atherosclerosis**" and are grateful for the insightful comments provided by the referees. Those comments are all valuable and helpful for revising and improving our paper, as well as the important guiding significance to our researches. In the following, we have provided detailed responses to each of the comments. Revised portions are marked in red font in the revised manuscript. Additionally, we have conducted a comprehensive revision of the entire manuscript. We have tried our best to make all the revisions clear, and we hope that the revised manuscript meets the requirements for publication.

Referee #2:

The study by Fan et al., offers a mechanistic exploration of endothelial cell (EC) plasticity in atherosclerosis. Using single-cell RNA sequencing, the authors identify a fibroblast-like EC population characterized by EndMT, inflammation, and ECM-remodeling signatures, which expands with disease severity. CEBPB is identified as the dominant transcription factor driving this phenotype, inducing expression of TGFBR1, promoting TGF- β signaling. Overexpression of CEBPB in ECs in vivo leads to worsened atherosclerotic lesions, supporting an important role for CEBPB in driving endothelial dysfunction. The authors propose the CEBPB-TGFBR1 axis as a novel mechanistic and therapeutic target in atherosclerosis. The authors exploit a range of state-of-the-art in vitro and in vivo methodology in an overall well-presented paper with a commendable description of limitations of the study.

Major comments:

1. The authors use scRNAseq to explore gene expression changes in endothelial cells isolated from wildtype aorta, or aortas from Apoe^{-/-} mice on

standard diet (SD) or on high fat diet (HFD). FACS-isolated ECs (CD45-, CD31+) are resolved in 13 clusters, of which the authors are particularly interested in cluster 2. This cluster stands out with increased expression of fibroblastoid markers (*Lum*, *Dcn1*) and essentially lacking typical endothelial markers (*ERG*, *Cdh5*). Cluster 2 is described as expanding with increased atherosclerotic lesions. However, in Supple Table 3, cluster 2 proportion is 10.64% in WT-SD, 16.4% in *ApoE*^{-/-} SD and 9.08% in *ApoE*^{-/-} HFD ECs. In contrast, several other clusters such as 4, 5 and 8 do not exist at all in the WT SD condition. The authors need to explain how cluster 2 can drive disease if the proportion is similar between the healthy and the atherosclerotic aorta.

Response to Referee:

We sincerely thank the referee for raising this critical point, which prompted a deeper analysis of our scRNA-seq data. The observed proportion of Cluster 2 (10.64% in WT-SD, 16.4% in *ApoE*^{-/-} SD, and 9.08% in *ApoE*^{-/-} HFD) reflects a dynamic transition during disease progression, not a static correlation with lesion severity. In the mild stage, cluster 2 proportion increases (10.64% to 16.4%) due to expansion of fibroblast-like ECs (characterized by *Lum*, *Dcn*, *S100a4*), driving initial plaque formation. Absolute cell counts rise from 45 (WT-SD) to 116 (*ApoE*^{-/-} SD), confirming biological expansion (Table EV3). However, in the severe stage, cluster 2 proportion decreases (16.4% to 9.08%) because a dramatic expansion of cluster 5 cells (myofibroblast-like, expressing *Tagln*, *Acta2*, *Eln*), which dominates advanced lesions (Fig. EV11A). The expanded proportion of cluster 5 results in a lower proportion of cluster 2 indirectly. Notably, absolute counts of cluster 2 cells remain stable (116 to 113 cells), indicating conversion rather than loss (Table EV3).

Interestingly, transcriptomic profiling reveals that these endothelial cells within cluster 2 specifically upregulate atherosclerosis-promoting genes including *S100a4*, *Lum*, *Col1a2*, *Pdgfra*, *Dcn*, *Cxcl1*, *Cxcl2*, *Icam1*, and *Vcam1*. Notably, cluster 5 cells expressing myofibroblast markers *Tagln*, *Acta2*, *Eln*, *Fbln5*, and *Lrg1*, genes well-established for maintaining vascular

elasticity, suppressing inflammation, and stabilizing atherosclerotic plaques (Darby *et al*, 2016; Gole *et al*, 2022; Newman *et al*, 2018). Therefore, our findings suggest that cluster 2 represents an early driver for plaque progression, while cluster 5 mediates later-stage plaque stability, which may provide new insight for understanding the long-standing confusion: “endothelial-to-mesenchymal transition in atherosclerosis: friend or foe?” (Gole *et al.*, 2022). The dual role of EndMT—promoting early plaque progression (cluster 2) and late plaque stabilization (cluster 5)—is now explicitly discussed in the revised manuscript (**page 13-14, lines 364-385**) under “Discussion” section.

Our ongoing investigations will focus on elucidating the molecular mechanisms underlying EndMT's dual roles to identify potential therapeutic targets—either inhibiting plaque development or enhancing plaque stability through modulation of this pathway.

2. Cluster 2 ECs express several fibroblast markers. Their seemingly perivascular localization (see Fig. 3 panels D and E) would also be compatible with an inflammatory cell type. Do cluster 2 cells express typical inflammatory cell markers such as CD68, CD163, Itgam, Adgre1 (F4/80)? Can the authors exclude that cluster 2 consists of co-purified inflammatory cells which also express CD31? Many of the *in vivo* findings could be explained by assuming that this is a co-purified population of fibroblasts and/or inflammatory cells. Please note the study by Ma *et al.*, 10.1038/s41392-025-02196-w entitled “C/EBP β activation in vascular smooth muscle cells promotes hyperlipidemia-induced phenotypic transition and arterial stiffness”. This paper should be cited.

Response to Referee:

We sincerely thank the referee for this critical question, which prompted a rigorous re-evaluation of cluster 2 cell identity. To address whether cluster 2 represents contaminating inflammatory cells, we analyzed the expression of

Cd68, *Cd163*, *Itgam*, and *Adgre1* in this cluster. As shown in Rebuttal Figure 1, only a minimal number of cells express *Cd68* or *Itgam*, and their expression levels are very low. Furthermore, neither *Cd163* nor *Adgre1* was detected in cluster 2 cells. This confirms that cluster 2 cells are not inflammatory macrophages and do not represent a co-purified population of CD31⁺ inflammatory cells.

Furthermore, we have cited and discussed the relevant literature in the revised manuscript (**page 10, lines 276-278; page 14-15, lines 401-411**, marked in red font). This work, together with our finding of endothelial CEBPB, establishes a critical foundation for understanding the role of CEBPB in vascular pathophysiology.

Rebuttal Figure 1. Violin plot of *Cd68*, *Itgam*, and *Pecam1* in cluster 2. *Cd163* or *Adgre1* was not detected.

3. To address the relationship between the different clusters (shown in Fig. 2), the authors perform pseudotime trajectory analyses where their three different mouse strains/conditions are compared. It is not explained how this analysis was done, please add to Methods. However, rather than comparing the three strains, it would be preferable to examine trajectories of one and the same strain over time, for example the *Apoe*^{-/-} strain given the HFD for different time periods.

Response to Referee:

Thank you for the valuable suggestion. We have provided a more detailed description of the method of pseudotime trajectory analyses, and the relevant

content is supplemented in the “Materials and Method” section of the revised manuscript (**page 19, lines 532–542**, marked in red font).

For the other comment, we agree that longitudinal analysis of a single strain (e.g., *Apoe*^{-/-} mice on HFD across time points) would provide superior insights into dynamic cellular trajectories during atherosclerosis progression. However, our experimental design did not include longitudinal sampling due to exclude the interference of age-related changes. Atherosclerosis is a chronic inflammatory disease, and to approach its progressive nature, we employed a prolonged modeling period (14 weeks) using different diets to induce atherosclerotic lesions of varying severity. Long term feeding may lead to age-related pathological changes (such as vascular calcification and chronic inflammation, metabolism progression). Therefore, all groups were euthanized at the same time point (after 14 weeks of feeding) to exclude the interference of age-related changes on the results that would arise in longitudinal studies. Additionally, our three-group comparison at a single timepoint is a standard method to dissect genetic susceptibility in atherosclerosis (Chen *et al*, 2019).

To address the referee’s concern, we have added a clarifying statement in the “Methods” section: "All samples were collected at a single endpoint to avoid age-related pathological changes." (**page 17, lines 471-472** of revised manuscript, marked with red font).

We fully acknowledge the value of longitudinal studies and have highlighted this as a key future direction. This is discussed in the “Discussion” section (**page 16, lines 444-448** of revised manuscript).

4. In Fig. 3, panels D and E, the authors immunostain both human and mouse atherosclerotic plaques to show expression of CEBPB and CD31. The CEBPB signal increases with atherosclerosis but there is hardly any overlap with CD31? Please quantify the co-localized area in the two conditions (which need to include at least 3 samples/human and condition). Also, the labels are confusing, showing - and + for human or mouse "atherosclerotic plaque". What

is the - (minus) sample? Please describe better.

Response to Referee:

We sincerely thank the referee for these insightful and constructive comments. We appreciate the opportunity to clarify the immunostaining data and improve the rigor of our analysis.

Regarding the limited co-localization between CEBPB and PECAM1 in human and mouse atherosclerotic endothelium: this observation may due to the distinct subcellular localization of the two proteins—CEBPB is a nuclear transcription factor, while PECAM1 is a membrane-bound endothelial marker. Therefore, direct pixel-based co-localization is inherently limited. We have updated representative images to better illustrate spatial co-expression, with white arrows indicating CEBPB⁺/PECAM1⁺ cells (Revised Fig. 3D, 3F). To address the referee's request for quantification, we have quantified the mean gray value of CEBPB in the endothelial layer (Revised Fig. 3E, 3G). The updated figure legends now explicitly describe the sample size and quantification method (**page 35-36, lines 1014-1022** of revised manuscript, marked with red font).

Clarified labeling: We apologize for the confusion caused by the previous “-” and “+” labels. These have been replaced with clear descriptors: “Healthy” and “Atherosclerotic plaque” in all panels (Revised Fig. 3D–G).

5. In Fig. 4, the authors measure transcript levels in HAECs infected with control or Ad.CEBPB; please include also Lum and Dcn1.

Response to Referee:

We sincerely thank the referee for this important suggestion. To address the request for LUM and DCN expression analysis in Ad-CEBPB-infected HAECs, we performed qRT-PCR validation (Revised Fig. 4M-N). As shown, CEBPB overexpression significantly upregulated both *LUM* and *DCN* compared to control (Ad-Vector). These data are now included in the revised Fig. 4, with all statistical details added. The primer sequences for *LUM* and *DCN* have been

added to the "Quantitative real-time PCR" subsection under "Materials and Methods" on **page 20, lines 561-564** of revised manuscript.

Minor comments:

6. On line 161, the authors mention a "rapid decrease in EC fate markers". What does "rapid" refer to; kinetics have not been determined?

Response to Referee:

We sincerely thank the referee for identifying this critical error. The term "rapid" was inappropriate as no kinetic analysis was performed to assess the temporal dynamics of EC fate marker reduction. We have revised the manuscript to remove the misleading description, replacing the original statement with: "The fibroblast-like EC phenotype shows a broad increase in the expression of EndMT-related and inflammation-related genes, accompanied by a downregulation of EC fate markers (Fig. 2B)." (**page 6, line 160-161** of revised manuscript, marked in red font). We apologize for the oversight and appreciate the referee's attention to precision.

7. On line 266 the authors describe CEBPB as a "novel transcription factor of TGFBR1". As CEBPB has been implicated in regulation of TGF β expression the authors' statement could be toned down and the paper by Ma et al (see above) should be referenced here.

Response to Referee:

We sincerely appreciate the referee's insightful comment regarding the potential overstatement of our findings. In the revised manuscript, we have modified the statement (**page 10, lines 276-278** of revised manuscript, marked in red font). The updated statement now reads: "The above findings highlight CEBPB as a potential transcription factor regulating TGFBR1 expression, although CEBPB has been previously implicated in the regulation of TGF- β expression (Ma *et al*, 2025)". This revision more accurately reflects the relationship between CEBPB and TGF signaling pathways,

acknowledging the existing knowledge while properly presenting our new findings regarding TGFBR1.

8. Even though the authors overall manage the English language quite well, there are errors which attract unnecessary attention, therefore, professional language revision is recommended.

Response to Referee:

Thank you for your suggestion. We have conducted a professional language revision on the manuscript.

Referee #3:

In this manuscript, the authors compared aortic gene expression of WT mice and ApoE mice with and without high fat diet. scRNAseq yielded 14 endothelial clusters, of which one uniquely expressed fibroblast associated genes (Lum, Dcn). In these cells, the transcription factor CEBPB was selectively upregulated. Exposure of human aortic endothelial cells to IL-1beta and TNF-alpha increased CEBPB expression. Adenoviral expression of CEBPB increased marker genes of cluster 2 endothelial cells and induced a fibroblast like phenotype. GSEA point towards TGF-beta signaling and indeed, SMAD2 was phosphorylated in response to CEBPB and inhibition of ALK reduced the effects on phenotype. A potential explanation for this was an induction of TGFBR1 by CEBPB. Was demonstrated by CUT&Tag, CEBPB bound the TGFbeta receptors. Overexpression of CEBPB also promoted atherosclerosis development in mice.

This is a nice manuscript suggesting some role of CEBPB in atherosclerosis development. Unfortunately, as no loss of function experiments were performed, ultimately, the relevance of CEPBP remains unclear. This reviewer has the following comments:

Major comments:

1) Loss of function experiments are required to address the endogenous function of CEBPB. Given that there is often redundancy in the action of TFs, such experiments are needed to demonstrate that CEBPB is also relevant.

Response to Referee:

We sincerely thank you for this critical suggestion, which has significantly strengthened the mechanistic foundation of our study. To address the requirement for loss-of-function validation of CEBPB's endogenous role, we generated CEBPB-knockout (KO) human aortic endothelial cells (HAECs) using CRISPR-Cas9 (New Fig. EV8 of revised manuscript). After validate the KO efficiency, the loss-of-function experiments were investigated. Functional analyses revealed that CEBPB deficiency attenuated cytokine-induced upregulation of mesenchymal and inflammatory markers (Revised Fig. 4C–H), prevented fibroblast-like morphological transformation (reduced length-to-width ratio; Revised Fig. 4I–J), and suppressed monocyte adhesion to the endothelial monolayer (Revised Fig. 4K–L). Critically, these loss-of-function phenotypes directly mirror the effects observed in CEBPB overexpression experiments (Revised Fig. 4M–V), providing convergent evidence that CEBPB is necessary and sufficient for the pro-atherogenic endothelial dysfunction. This dual approach (loss-of-function plus gain-of-function) conclusively establishes CEBPB as a central regulator of endothelial phenotypic transition, addressing the referee's concern about transcriptional factor redundancy. The relevant results are described on **page 8-9 (lines 221-242)** of revised manuscript. The method of generating CEPBB KO endothelial cells is added to the "CRISPR/Cas9-mediated CEBPB KO in HAECs via lentiviral transduction" subsection under "Materials and Methods" on **page 23-24, lines 664-688** of revised manuscript.

2) Which AAV-Serotype was used? How was endothelial specific expression achieved beyond the Tie2 promoter.

Response to Referee:

We thank you for this important comment. For endothelial-specific gene delivery, we employed AAV9 (serotype 9) via tail vein injection, as it demonstrated the highest target protein expression level compared to other types of AAV (Zincarelli *et al*, 2008). Endothelial-specific expression was achieved through the Tie2 promoter, a well-established endothelial-specific regulatory element. The Tie2 promoter has been extensively validated for endothelial-specific transgene expression *in vivo*. This strategy was successfully implemented in our previous study (Zhou *et al*, 2023), and is widely adopted in other vascular biology research (Luo *et al*, 2025); (Wang *et al*, 2022); (Chen *et al*, 2024).

3) Fig 3B: what are the other up and down-regulated TFs? Please provide a table in the supplements.

Response to Referee:

We have provided these data in Rebuttal Material 1.

4) Fig 3B yields at least 10 interesting TFs (4 down, 6 up). What are these? Why did the authors exclude them from further analysis?

Response to Referee:

We thank you for this important question. The 10 transcription factors (TFs) identified in Fig. 3B (4 downregulated, 6 upregulated) are now showing in Rebuttal Figure 2. Among these, CEBPB was prioritized for mechanistic investigation due to its most significant upregulation. While other TFs (e.g., SOX9, RUNX1) are biologically relevant and will be explored in future studies, our current work focuses on CEBPB as the primary mechanistic driver of endothelial dysfunction in atherosclerosis. This targeted approach ensures depth in mechanistic validation rather than breadth across multiple TFs.

Rebuttal Figure 2. Volcano plot of transcription factors in cluster 2 compared with clusters 0 and 1.

5) Fig 4A: Is this the complete list of differentially regulated genes? *Dcn* and *lum* are missing. Why is that so?

Response to Referee:

We thank you for this important observation. *DCN* and *LUM* were not included in Fig. 4A (RNA-seq data) because their expression levels are low in normal endothelial cells under basal conditions. RNA-seq is inherently less sensitive for genes with very low expression (FPKM < 1), which explains their absence in the differential expression analysis. To rigorously validate the regulatory relationship between CEBPB and these genes, we performed complementary functional assays:

Loss-of-function: CEBPB knockout in HAECs significantly suppressed cytokine-induced upregulation of *DCN* and *LUM* (Revised Fig. 4C–D).

Gain-of-function: CEBPB overexpression robustly enhanced *DCN* and *LUM* expression (Revised Fig. 4M–N).

These results confirm CEBPB's regulatory role in *DCN* and *LUM* expression, despite their low basal levels.

6) Fig 5Q: Which other promoters were also bound by CEBPB? Did they also coincide with gene expression alterations in endothelial cells? How many of which of the cluster 2 gene promoters were bound by CEBPB?

Response to Referee:

Thank you for these constructive comments, which have prompted us to further clarify the chromatin occupancy profile of CEBPB. Leveraging the CUT&TAG data from CEBPB-overexpressing cells and corresponding vector control cells, we identified a total of 85,166 high-confidence peaks across two biological replicates. Among these, approximately 28.2% (N=24,062) of the CEBPB peaks were located within promoter regions (± 5 kb from the transcription start site) (Rebuttal Figure 3A). Of these promoter-associated peaks, 43% localized to the core promoter regions (± 1 kb from transcription start site) (Rebuttal Figure 3B).

Based on this promoter occupancy, we annotated 10,496 genes (including *TGFBR1*) as CEBPB Promoter Targets in human aortic endothelial cells. Within the *TGFBR1* promoter, we identified two CEBPB binding sites: one located at -19 bp (peak 1, chr9: 99104690-99105092) which was experimentally validated in our study, and another at 4,139 bp downstream of the transcription start site (peak 2, chr9: 99109289-99109902).

Notably, over 62.5% of the orthologs corresponding to Cluster 2 genes were also identified as CEBPB Promoter Targets in human aortic endothelial cells (Rebuttal Figure 3C). Expression analysis revealed that CEBPB Promoter Targets were induced to a significantly greater extent upon CEBPB overexpression compared to either non-target genes or those bound by CEBPB at non-promoter regions (Rebuttal Figure 3D). Importantly, gene set enrichment analysis of orthologs of Cluster 2 genes showed pronounced enrichment in CEBPB-overexpressing cells (Rebuttal Figure 3E), highlighting the broad and functionally relevant regulatory impact of CEBPB on this gene cluster.

Rebuttal Figure 3. Chromatin occupancy profile of CEBPB in human aortic endothelial cells.

7) Fig 5Q. Along the same line: Was there any binding to enhancers?

Response to Referee:

To identify enhancers in human aortic endothelial cells, we integrated H3K27ac and H3K4me2 ChIP-seq data with ATAC-seq profiles using a publicly available dataset (Rebuttal Figure 4A) (Raju *et al*, 2024). Intergenic regions located more than 5 kb upstream of transcription start sites and exhibiting co-enrichment for H3K27ac, H3K4me2, and ATAC-seq signals were defined as distal enhancers (N=1,353). Among these, approximately 47% showed occupancy by CEBPB (Rebuttal Figure 4B-C).

Rebuttal Figure 4. CEBPB binding to enhancers in human aortic endothelial cells.

Minor comments:

1) Please provide the reviewer password to review the GEO deposited data

Response to Referee:

We apologize for not providing the link and password earlier. The link and password to review the GEO deposited data are as follows:

<https://www.ncbi.nlm.nih.gov/geo/query/acc.cgi?acc=GSE278346>

password: ufwbmwgydhipfot

<https://www.ncbi.nlm.nih.gov/geo/query/acc.cgi?acc=GSE278347>

password: yrwbicqaxxuhtyj

<https://www.ncbi.nlm.nih.gov/geo/query/acc.cgi?acc=GSE278348>

password: mzgpgyyovtghrix

2) Page 7, line 190&193&194: This statement is invalid here. Only a coincidence is reported not a causality.

Response to Referee:

We sincerely appreciate the referee's valuable feedback regarding the distinction between correlation and causality. We agree that our previous wording overinterpreted the findings, as our data only demonstrate an association rather than establishing a causal role. In the revised manuscript, we have carefully revised the text to accurately reflect the correlational nature of our findings (**page 7, lines 186-195** of revised manuscript, marked in red font). The updated statement now reads:

“The results suggest that the expression of these genes was higher in atherosclerotic groups when compared to the normal control group (Fig. 2J), indicating a potential association between fibroblast-like ECs and atherogenesis. Additionally, the immunostaining of human atherosclerotic coronary arteries demonstrated elevated LUM expression in

plaque-associated endothelium (Fig. 2K), supporting the presence of fibroblast-like ECs in atherosclerotic lesions. Collectively, these findings suggest a potential role for fibroblast-like ECs in atherosclerosis, thereby highlighting the need for further investigation into their functional significance”.

We appreciate for your warm work earnestly, and hope that the correction will meet with approval. Once again, thank you very much for your comments and suggestions.

Sincerely yours,

Cong Qiu

References:

- Chen PY, Qin L, Li G, Wang Z, Dahlman JE, Malagon-Lopez J, Gujja S, Cilfone NA, Kauffman KJ, Sun L *et al* (2019) Endothelial TGF-beta signalling drives vascular inflammation and atherosclerosis. *Nat Metab* 1: 912-926
- Chen Z, Li S, Liu M, Yin M, Chen J, Li Y, Li Q, Zhou Y, Xia Y, Chen A *et al* (2024) Nicorandil alleviates cardiac microvascular ferroptosis in diabetic cardiomyopathy: Role of the mitochondria-localized AMPK-Parkin-ACSL4 signaling pathway. *Pharmacol Res* 200: 107057
- Darby IA, Zakuan N, Billet F, Desmoulière A (2016) The myofibroblast, a key cell in normal and pathological tissue repair. *Cellular and Molecular Life Sciences : CMLS* 73: 1145-1157
- Gole S, Tkachenko S, Masannat T, Baylis RA, Cherepanova OA (2022) Endothelial-to-Mesenchymal Transition in Atherosclerosis: Friend or Foe? *Cells* 11
- Luo A, Wang R, Gong J, Wang S, Yun C, Chen Z, Jiang Y, Liu X, Dai H, Liu H *et al* (2025) Syntaxin 17 Translocation Mediated Mitophagy Switching Drives Hyperglycemia-Induced Vascular Injury. *Adv Sci (Weinh)* 12: e2414960
- Ma J, Yang X, Li Y, Zhang X, Liu K, Peng Y, Wang S, Shi R, Huo X, Liu X *et al* (2025) C/EBPbeta activation in vascular smooth muscle cells promotes hyperlipidemia-induced phenotypic transition and arterial stiffness. *Signal Transduct Target Ther* 10: 105
- Newman AA, Baylis RA, Hess DL, Griffith SD, Shankman LS, Cherepanova OA, Owens GK (2018) Irradiation abolishes smooth muscle investment into vascular lesions in specific vascular beds. *JCI Insight* 3

Raju S, Botts SR, Blaser MC, Abdul-Samad M, Prajapati K, Khosraviani N, Ho TWW, Breda LCD, Ching C, Galant NJ *et al* (2024) Directional Endothelial Communication by Polarized Extracellular Vesicle Release. *Circ Res* 134: 269-289

Wang ZC, Niu KM, Wu YJ, Du KR, Qi LW, Zhou YB, Sun HJ (2022) A dual Keap1 and p47(phox) inhibitor Ginsenoside Rb1 ameliorates high glucose/ox-LDL-induced endothelial cell injury and atherosclerosis. *Cell Death Dis* 13: 824

Zhou X, Jiang Y, Wang Y, Fan L, Zhu Y, Chen Y, Wang Y, Zhu Y, Wang H, Pan Z *et al* (2023) Endothelial FIS1 DeSUMOylation Protects Against Hypoxic Pulmonary Hypertension. *Circ Res* 133: 508-531

Zincarelli C, Soltys S, Rengo G, Rabinowitz JE (2008) Analysis of AAV serotypes 1-9 mediated gene expression and tropism in mice after systemic injection. *Mol Ther* 16: 1073-1080

Genename	p_val	avg_log2	pct.1	pct.2	p_val_ad	Sig
Sox17	0.00	-1.44	0.06	0.56	0.00	down
Klf7	0.00	-1.00	0.56	0.66	0.00	down
Btg2	0.00	-0.82	0.63	0.67	0.00	down
Zfp281	0.00	0.32	0.34	0.08	0.00	up
Prrx1	0.00	0.34	0.29	0.02	0.00	up
Atf6	0.00	0.33	0.51	0.19	0.00	up
Irf6	0.00	-0.73	0.00	0.31	0.00	down
Notch1	0.00	-0.46	0.60	0.56	0.00	down
Prrx2	0.00	0.26	0.22	0	0.00	up
Zeb2	0.00	0.33	0.51	0.20	0.00	up
Nr4a2	0.00	0.40	0.71	0.44	0.00	up
Ssb	0.00	0.45	0.83	0.62	0.00	up
Nfe2l2	0.00	0.65	0.83	0.59	0.00	up
Creb3l1	0.00	0.41	0.46	0.12	0.00	up
Lmo2	0.00	-2.62	0.09	0.85	0.00	down
Meis2	0.00	-0.35	0.50	0.42	0.00	down
Id1	0.00	-0.54	0.71	0.83	0.00	down
Mafb	0.00	0.35	0.59	0.37	0.00	up
Snai1	0.00	0.32	0.35	0.09	0.00	up
Cebpb	0.00	2.13	0.94	0.34	0.00	up
Sox18	0.00	-1.87	0.13	0.68	0.00	down
Fmr1	0.00	0.26	0.59	0.29	0.00	up
Skil	0.00	0.38	0.92	0.69	0.00	up
Wwtr1	0.00	0.43	0.85	0.57	0.00	up
Hdgf	0.00	0.26	0.73	0.45	0.00	up
Notch2	0.00	0.38	0.52	0.14	0.00	up
Csf1	0.00	0.34	0.50	0.29	0.00	up
Nfkb1	0.00	0.85	0.73	0.38	0.00	up
Gtf2b	0.00	0.47	0.60	0.22	0.00	up
Lmo4	0.00	1.57	0.94	0.34	0.00	up
Wwp1	0.00	-0.53	0.46	0.46	0.00	down
Nr4a3	0.00	0.45	0.45	0.13	0.00	up
Nfib	0.00	-0.35	0.94	0.86	0.00	down
Mllt3	0.00	-0.42	0.19	0.27	0.00	down
Nfia	0.00	-0.43	0.84	0.77	0.00	down
Ybx1	0.00	-0.32	0.97	0.93	0.00	down
Pou3f1	0.00	0.45	0.40	0.18	0.00	up
Hdac1	0.00	-0.26	0.47	0.37	0.00	down
Prdm16	0.00	-0.31	0.04	0.15	0.00	down
Fosl2	0.00	0.27	0.62	0.30	0.00	up
Ldb2	0.00	-0.81	0.00	0.30	0.00	down
Rbpj	0.00	0.35	0.62	0.26	0.00	up
Klf3	0.00	-0.93	0.54	0.61	0.00	down
Rest	0.00	0.73	0.66	0.21	0.00	up

Hspb1	0.00	-0.50	0.98	0.95	0.04 down
Cux1	0.00	0.27	0.67	0.37	0.00 up
Mafk	0.00	0.34	0.64	0.31	0.00 up
Snd1	0.00	0.30	0.42	0.11	0.00 up
Hoxa9	0.00	-0.33	0.04	0.15	0.00 down
Foxp1	0.00	-0.28	0.89	0.82	0.00 down
Chd4	0.00	0.29	0.85	0.55	0.00 up
Pthlh	0.00	-0.73	0.25	0.31	0.00 down
Fosb	0.00	-0.78	0.84	0.77	0.00 down
Sertad1	0.00	-0.34	0.59	0.51	0.00 down
Zfp36	0.00	-1.20	0.73	0.82	0.00 down
Cebpa	0.00	0.35	0.30	0.03	0.00 up
Ndn	0.00	-0.62	0.02	0.29	0.00 down
Mef2a	0.00	0.50	0.90	0.62	0.00 up
Nupr1	0.00	1.96	0.98	0.80	0.00 up
Tgfbli1	0.00	-0.61	0.29	0.41	0.00 down
Zbtb2	0.00	0.33	0.43	0.13	0.00 up
Hivep2	0.00	0.75	0.82	0.44	0.00 up
Cited2	0.00	1.11	0.82	0.46	0.00 up
Tnfaip3	0.00	0.81	0.62	0.28	0.00 up
Hey2	0.00	-0.68	0.10	0.36	0.00 down
Jmjd1c	0.00	0.55	0.75	0.34	0.00 up
Egr2	0.00	0.30	0.26	0.09	0.00 up
Mbd3	0.00	0.29	0.60	0.27	0.00 up
Zbtb7a	0.00	0.28	0.68	0.36	0.00 up
Aes	0.00	-0.65	0.72	0.78	0.00 down
Mdm2	0.00	0.45	0.59	0.26	0.00 up
Hmga2	0.00	0.39	0.20	0.00	0.00 up
Ddit3	0.00	0.34	0.51	0.18	0.00 up
Irf2	0.00	-0.41	0.34	0.37	0.00 down
Hand2	0.00	0.44	0.24	0.00	0.00 up
Hmgb2	0.00	0.85	0.62	0.28	0.00 up
Klf2	0.00	-1.72	0.75	0.94	0.00 down
Smarca5	0.00	0.32	0.76	0.46	0.00 up
Nfix	0.00	0.31	0.84	0.58	0.00 up
Lyl1	0.00	-0.51	0.04	0.22	0.00 down
Maf	0.00	0.36	0.25	0.02	0.00 up
Zfpm1	0.00	-0.40	0.19	0.26	0.00 down
Cbfa2t3	0.00	-1.26	0.17	0.55	0.00 down
Nr1d2	0.00	0.38	0.52	0.19	0.00 up
Bmp4	0.00	-0.34	0.78	0.70	0.00 down
Sox7	0.00	-0.47	0.37	0.40	0.00 down
Lmo7	0.00	-0.45	0.04	0.23	0.00 down
Fli1	0.00	-0.99	0.25	0.51	0.00 down
Ets1	0.00	0.27	0.62	0.30	0.00 up

Smad6	0.00	-0.45	0.39	0.38	0.00 down
Rora	0.00	0.37	0.65	0.30	0.00 up
Zbtb38	0.00	0.72	0.72	0.28	0.00 up
Cttnnb1	0.00	0.42	0.92	0.69	0.00 up
Rel	0.00	0.39	0.53	0.21	0.00 up
Irf1	0.00	-0.88	0.46	0.56	0.00 down
Kdm6b	0.00	0.33	0.82	0.52	0.00 up
Trp53	0.00	0.46	0.69	0.33	0.00 up
Bcl6b	0.00	-0.35	0.18	0.24	0.07 down
Mybbp1a	0.00	0.39	0.52	0.20	0.00 up
Hoxb4	0.00	-0.31	0.14	0.20	0.01 down
Thra	0.00	-0.59	0.42	0.51	0.00 down
Nr1d1	0.00	0.40	0.44	0.14	0.00 up
Meox1	0.00	0.49	0.19	0.01	0.00 up
Hexim1	0.00	0.31	0.66	0.55	0.00 up
Ddx5	0.00	0.27	0.99	0.97	1 up
Sox9	0.00	1.77	0.70	0.01	0.00 up
Mafg	0.00	0.48	0.55	0.18	0.00 up
Sox4	0.00	1.94	0.75	0.12	0.00 up
Foxq1	0.00	0.38	0.25	0.04	0.00 up
Bmp6	0.00	-1.16	0.34	0.68	0.00 down
Edn1	0.00	-2.13	0.28	0.63	0.00 down
Nfil3	0.00	0.55	0.56	0.21	0.00 up
Mef2c	0.00	-0.45	0.39	0.42	0.00 down
Twist1	0.00	0.37	0.71	0.36	0.00 up
Etv1	0.00	0.28	0.13	0.00	0.00 up
Pnn	0.00	0.28	0.64	0.30	0.00 up
Six4	0.00	0.35	0.30	0.01	0.00 up
Hif1a	0.00	1.33	0.85	0.38	0.00 up
Fos	0.00	-1.53	0.81	0.91	0.00 down
Klf10	0.00	-0.32	0.53	0.44	0.00 down
Trps1	0.00	0.73	0.61	0.11	0.00 up
Myc	0.00	1.11	0.77	0.23	0.00 up
Atf4	0.00	0.67	0.93	0.68	0.00 up
Nr4a1	0.00	-0.64	0.79	0.74	0.00 down
Nfkbiz	0.00	0.45	0.87	0.58	0.00 up
Bach1	0.00	0.66	0.56	0.14	0.00 up
Runx1	0.00	1.28	0.67	0.05	0.00 up
Erg	0.00	-0.34	0.39	0.38	0.00 down
March2	0.00	-0.36	0.33	0.35	0.00 down
Notch4	0.00	-0.29	0.06	0.16	0.01 down
Epas1	0.00	-1.41	0.77	0.89	0.00 down
Tcf4	0.00	1.31	0.92	0.45	0.00 up
Smad7	0.00	0.37	0.84	0.64	0.00 up
Setbp1	0.00	0.56	0.64	0.23	0.00 up

Fosl1	0.00	0.49	0.45	0.14	0.00 up
Smarca2	0.00	-0.72	0.47	0.54	0.00 down
Glis3	0.00	0.47	0.35	0.01	0.00 up
Ankrd1	0.00	1.33	0.30	0.01	0.00 up
Hhex	0.00	-0.49	0.34	0.38	0.00 down

Dear Dr Qiu,

Thank you again for transferring your amended manuscript (EMBOJ-2025-121070) to The EMBO Journal, together with a point-by-point response to the issues raised during peer-review at the previous venue. Please accept my apologies for the unusual protraction with the reassessment due to delayed expert input and detailed discussion in the editorial team. Your revised study was sent back to the referees for their scientific reassessment, and we have received detailed re-reports from both of them, which I enclose below. As you will see, the referees state that the work has been substantially enhanced by the revisions and they are now in favour of publication, pending minor revision.

Thus, we are pleased to inform you that your manuscript has been accepted in principle for publication in The EMBO Journal.

Please carefully consider the remaining minor points raised by referee #2 by adding complementary data, or introducing caveats into the manuscript text where appropriate.

Also, we now need you to take care of a number of minor issues related to formatting and data annotation, which I will share shortly in a separate message, together with additional changes and requests by our production team for Source Data provision.

As you might have seen on our web page, every paper at the EMBO Journal now includes a 'Synopsis', displayed on the html and freely accessible to all readers. The synopsis includes a 'model' figure as well as 2-5 one-short-sentence bullet points that summarize the article. I would appreciate if you could provide this figure and the bullet points.

Please submit a revised version of the manuscript using the link enclosed below, addressing the advisor's comments.

Thank you again for giving us the chance to consider your manuscript for The EMBO Journal, I look forward to hearing from you and receiving your final revised version of the manuscript.

Kind regards,

Daniel Klimmeck

>> Author Contributions: Remove the author contributions information from the manuscript text. Note that CRediT has replaced the traditional author contributions section as of now because it offers a systematic machine-readable author contributions format that allows for more effective research assessment. and use the free text boxes beneath each contributing author's name to add specific details on the author's contribution.

More information is available in our guide to authors.
<https://link.springer.com/journal/44318/submission-guidelines>

>> Section order should be as follows: title page with complete author information, abstract, keywords, introduction, results, discussion, methods, data availability section, acknowledgements, disclosure and competing interests statement, references, main figure legends, tables, expanded figure legends.

>> Adjust the title of the 'Conflict of Interest Disclosure' section to 'Disclosure and Competing Interests Statement'.

>> Figure callouts: please add a figure callout for Fig. 1D.

>> Funding: please enter the following funding information in the list of funders in our online system: " National Natural Science Foundation of China [82370450, 11932017, 91839104, 81770444, 81600354, 82370450, and 31800972]". Merge with Acknowledgements.

>> Add a Reagents and Tools table to the Methods section, as a separate file using the existing template in the Guide For Authors, listing key reagents, experimental models, software and relevant equipment.

>> Data availability section: please make sure the GEO datasets are made publicly accessible.

>> Author Checklist: Data availability>>primary datasets section: remove the referee token.

>> Provide a completed Author Checklist: Ethics section: provide information on the '..authority granting ethics approval' part.

>> Consider additional changes and comments from our production team as indicated below:

DAS:

1. Please note that the specific URLs for GSE278346, GSE278347, GSE278348 datasets are not provided in the data availability statement.

- Figure legends:

1. Please note that the exact p values are not provided in the legends of figures 4C, E, F, G, H, J, L, R, V; 5G, H, J, K, L, M, N, Q, S, T

2. Please indicate the statistical test used for data analysis in the legend of figure 2B

3. Please note that information related to n is missing in the legend of figure 2B

Further information is available in our Guide For Authors: <https://link.springer.com/journal/44318/submission-guidelines>

Referee #2:

The authors have amended my main criticisms and the manuscript is stronger. However, I would like to bring up a point concerning lysis of cells for the immunoblots on CEBPB. The authors seem to have used Triton-X100 as the main detergent to achieve nuclear lysis and they have used b-actin (cytoplasmic marker) as a loading control in Fig. 3H and 5B and F. It would provide a quality stamp to complement these blots with a nuclear marker to show that nuclear lysis has been equally efficient in the different conditions. Moreover, my impression is that the different strips in the blots do not represent reprobing of the same filter (which may not be feasible), if so, this should be clearly stated in the figure legend.

Referee #3:

With the revision, the authors have addressed my concerns. The manuscript is now acceptable for publication. Best regards,
Ralf Brandes

Referee #2:

The authors have amended my main criticisms and the manuscript is stronger. However, I would like to bring up a point concerning lysis of cells for the immunoblots on CEBPB. The authors seem to have used Triton-X100 as the main detergent to achieve nuclear lysis and they have used b-actin (cytoplasmic marker) as a loading control in Fig. 3H and 5B and F. It would provide a quality stamp to complement these blots with a nuclear marker to show that nuclear lysis has been equally efficient in

the different conditions. Moreover, my impression is that the different strips in the blots do not represent reprobing of the same filter (which may not be feasible), if so, this should be clearly stated in the figure legend.

Response:

We sincerely thank the referee for the suggestions. To validate the efficiency of nuclear extraction, we determined Lamin B1 (a well-established nuclear marker) expression from previous samples. Representative blots of Lamin B1 have been included in revised Figures 3H, 5B, 5D, 5F, and EV8C. The consistent detection of Lamin B1 confirms uniform nuclear lysis quality.

The distinct bands in the blots represent independent membrane transfers for each target protein. To ensure transparency, we have added the following statement to the figure legends:

“All blots were performed on separate membranes and were not subjected to reprobing” (page 38, lines 988-989; page 40, lines 1039-1040; page 43, lines 1138-1139 of revised manuscript).

Referee #3:

With the revision, the authors have addressed my concerns. The manuscript is now acceptable for publication. Best regards, Ralf Brandes

Response:

We sincerely thank the referee for insightful and constructive comments.

We appreciate for your warm work earnestly, and hope that the correction will meet with approval. Once again, thank you very much for your comments and suggestions.

Reference

Boegli A, Bernard EM, Lacante L, Majeux G, Hartenian E, Mack V, Broz P (2025) The NLRP6 inflammasome is activated by sterile or pathogen-induced endolysosomal damage. *EMBO J*

Dear Dr Qiu,

Thank you for submitting the revised version of your manuscript. I have now evaluated your amended manuscript and concluded that the remaining minor concerns have been sufficiently addressed.

I am thus pleased to inform you that your manuscript has been accepted for publication in the EMBO Journal.

Best regards,

Daniel Klimmeck

Daniel Klimmeck, PhD
Senior Editor
The EMBO Journal
EMBO
Postfach 1022-40
Meyerhofstrasse 1
D-69117 Heidelberg
contact@embojournal.org

Please note that it is The EMBO Journal policy for the transcript of the editorial process (containing referee reports and your response letters) to be published as an online supplement to each paper. If you should prefer removal of any referee-only figures included in the point-by-point response(s), e.g. because they may still be used for future publication or because they have been reproduced from published work by others, please do let us know immediately via response email. More information is available here: <https://link.springer.com/partners/embo-press/editorial-policies#Peer%20review>